# From leaf to soil: *n*-alkane signal preservation, despite degradation along an environmental gradient in the tropical Andes.

Milan L. Teunissen van Manen[1], Boris Jansen[1], Francisco Cuesta[2], Susana León-Yánez[3], William D. Gosling[1].

[1] Institute for Biodiversity and Ecosystem Dynamics (IBED), University of Amsterdam, Amsterdam, The Netherlands.

[2] Grupo de Investigación en Biodiversidad, Medio Ambiente y Salud (BIOMAS), Universidad de Las Américas (UDLA), Quito, Ecuador.

[3] Escuela de las Ciencias Biológicas, Pontificia Universidad de Católica de Ecuador (PUCE), Quito, Ecuador.

*Correspondence to*: M.L. Teunissen van Manen (M.L.TeunissenvanManen@uva.nl)

**Abstract.** The relative abundance of *n*-alkanes of different chain lengths obtained from ancient soils and sediments have been used to reconstruct past environmental changes. However, interpretation of ancient *n*-alkane patterns relies primarily on modern plant wax *n*-alkane patterns measured from leaves. Little is still known about how *n*-alkane patterns, and environmental information therein, might be altered during the process of transfer from leaves into soil. We studied the *n*-alkane patterns extracted from leaves, necromass, and soil samples from an altitudinal gradient in the tropical Andes to clarify if the *n*-alkane pattern, and the local environmental information reflected, is altered as the plant source material degrades. We considered the (dis)similarity between *n*-alkane patterns in soil, necromass and leaves and specifically explored whether a temperature and/or precipitation signal is reflected in their *n*-alkane patterns. The *n*-alkane patterns showed degradation in soil as reflected by reduced carbon preference index (CPI). The lower CPI in soils as compared to leaves and necromass was significantly correlated with temperature and precipitation along the transect, most likely because of increased microbial activity under warmer and wetter conditions. Despite degradation, all samples types showed a systematic shift in longer vs. shorter *n*-alkanes when moving up the transect. Further examination revealed the systematic shift correlated with transect temperature and precipitation. Since transect vegetation is constant along the transect, this would appear to indicate the recording of a climatic signal within the *n*-alkane patterns that is preserved in the soil, albeit that the correlation was weaker there. The study results warrant further research into a possible underlying causal relationship that may lead to the development of *n*-alkanes patterns as a novel palaeoecological proxy.

## 1 Introduction

Making accurate reconstructions of past environments is important and urgent, as these can inform how modern environments could respond to current climatic and land use changes (Cronin, 2014). Interpretations of past environmental change are commonly obtained from fossil pollen, charcoal, and molecular proxies extracted from sediments and soils (Smol et al., 2001). To help improve these reconstructions of past environments and environmental change, the development of molecular proxies such as plant wax *n*-alkanes has seen much research attention over the last two decades (Jansen and Wiesenberg, 2017).

Plant wax *n*-alkanes (typically between $C_{21}$-$C_{35}$) are part of the protective outer wax layer of plants (Eglinton and Hamilton, 1967), and have been observed to reflect the local environmental conditions in which the plant is situated (for example, Feakins et al., 2016; Hoffmann et al., 2013; Teunissen van Manen et al., 2019). The *n*-alkanes produced by plants transfer to the soil and sedimentary records, where they are preserved and from which they can be extracted (Jansen and Wiesenberg, 2017). The retention of fossil *n*-alkanes in soils and sediments make *n*-alkanes a promising proxy to help reconstruct environmental change (Bush and McInerney, 2015; Hoffmann et al., 2013; Teunissen van Manen et al., 2019).

Multiple studies have already used plant wax *n*-alkane biomarkers to reconstruct past environmental change, mostly via shifts in plant community composition that are reflected in the *n*-alkane patterns, i.e. the relative abundance of a suite of *n*-alkanes of different chain-lengths (e.g. Crausbay et al., 2014; Jansen et al., 2013). For example, Crausbay et al. (2014) reconstructed past drought frequency in a tropical montane cloud forest on the Haleakalā volcano (Maui, Hawaii, USA) using *n*-alkane patterns. Specifically, Crausbay et al. (2014) calibrated the fossil *n*-alkane record using the modern *n*-alkane signal of the *Metrosideros polymorpha* species, which has a pubescent and a glabrous variety. The modern pubescent and glabrous *n*-alkane calibration data were then used to reconstruct past abundance of pubescent *vs.* glabrous plant varieties and, from this infer relatively wet and dry conditions (Crausbay et al., 2014). In the Crausbay et al. (2014) study, and all other studies using *n*-alkanes biomarkers, the inferred past environmental changes from the ancient *n*-alkane record are reliant on what is known about modern plant wax *n*-alkanes and how they reflect the environment from which they were sampled. However, most of the studies on environmental controls of plant wax *n*-alkanes have been done on leaf material, often fresh from the tree. While there are several studies that have examined degradation of leaf wax *n*-alkanes, often only a part of the leaves-necromass-soil chain was considered (e.g. Zech et al., 2011). As a result, our current understanding of the taphonomy of leaf wax derived *n*-alkane patterns is based on a handful of studies that considered the entire leaves-necromass-soil chain (e.g. Bush and McInerney, 2015; Howard et al., 2018; Schäfer et al., 2016; Tipple and Pagani, 2013; Wu et al., 2019).

As the source (plant) material degrades, in particular it is poorly understood to what extent the *n*-alkane patterns and the environmental information potentially contained therein are altered. For instance, it is generally accepted that the odd-over-even chain-length predominance of *n*-alkanes of higher chain-lengths as produced by higher plants, is often reduced during the transfer via litter to the soil or sediment (e.g. Schäfer et al., 2016). However, this does not affect the environmental information contained in the relative abundance of the odd carbon number *n*-alkanes as long as their relative distribution pattern remains the same (e.g. Jansen et al., 2008; Schäfer et al., 2016). This lack of knowledge on the taphonomy of the *n*-alkane signal and, particularly its influence on the information contained therein, means that our interpretations of the ancient *n*-alkane biomarker signal could be confounded by an unobserved bias caused by degradation processes (Wu et al., 2019). This could potentially lead to erroneous or biased inferences of past environmental change based on the *n*-alkane biomarker signal. Therefore, expanding our knowledge on the taphonomy of the *n*-alkane signal is key to reliably deploy plant wax *n*-alkane biomarkers to reconstruct past environmental change (Wu et al., 2019).

The study aim was to expand the knowledge on the taphonomy of the plant wax *n*-alkane signal by comparing the *n*-alkane patterns extracted from leaves, necromass and soils, sampled along an attitudinally driven environmental (temperature and precipitation) gradient in the tropical Andes (Ecuador. By choosing this particular gradient for our study we can build on the body of knowledge available on the transect (Pinto et al., 2018) and simultaneously elaborate on the *n*-alkane work previously done by us (Teunissen van Manen et al., 2019). Specifically, in this study we address: (1) to what extent the *n*-alkane pattern degrades as the source material degrades (i.e. to what extent are the *n*-alkane patterns found in leaves, necromass and soils similar?), and (2) to what extent the *n*-alkane patterns in leaves, necromass and soil show a similar systematic variation along the transect (i.e. do the *n*-alkane patterns found in leaves, necromass and soils reflect the environment similarly?). In the context of the second question we also explored if any systematic shift in *n*-alkane patterns along the transect can tentatively be related to the shifts in temperature and precipitation along the transect. We discuss our findings in light of the applicability and interpretation of the *n*-alkane biomarker records as a proxy for local environmental change in the past.

## 2 Materials and methods

### 2.1 Study site and sampling

This study used the Pichincha long-term forest development and carbon monitoring transect, situated on the north western flank of the Ecuadorian Andes (Fig. 1). The sampled transect was established in 2015 by the research non-profit organization Consorcio para el Desarrollo Sostenible de la Ecorregión Andina (CONDESAN) (http://condesan-ecoandes.org/), who catalogued the tree community composition and recorded environmental data at each permanent plot (hereafter 'plot') (Pinto et al., 2018; Pinto and Cuesta, 2019).

The extensive elevation gradient captured by the transect, between 632 and 3507 m a.s.l. (above sea level), induces multiple environmental gradients, among which temperature, humidity and precipitation gradients. Mean annual temperature and mean annual relative air humidity ('temperature' and 'humidity' hereafter) were calculated based on hourly data collected at each plot between 2016-2018 (see Teunissen van Manen et al. (2019) for more details). We obtained the plot annual precipitation (hereafter 'precipitation') from the CHELSA dataset at 30S resolution (1km) (Karger et al., 2017). In total, the transect temperature gradient spans 7.2- 21.6 °C, the precipitation gradient spans 1580 - 2448 mm and the humidity gradient spans 96.1 - 99.8%. Given that the variation in the latter is so small, the humidity gradient was not considered as a driving factor in this study. However, it was included as a separate factor in the correlation analysis to disentangle its influence from that of temperature and precipitation so that the overall level of noise is reduced.

All samples were collected from the Pichincha transect (Table 1). Each plot (60x60 m) was subdivided in 9 subplots of 20x20m. The leaf samples were collected and analysed in a previous study (Teunissen van Manen et al., 2019). For that study we

targeted two widely distributed genera along the transect, the *Miconia* and *Guarea* genera (see Teunissen van Manen et al. (2019) for the genus distributions along transect) (Pinto et al., 2018). From each individual we took one sample, which consisted of 20-25 leaf(lets) taken from the canopy. In total we collected 87 leaf samples from 14 plots (Table 1). For this study these samples were supplemented with new necromass and soil samples.

Necromass samples were collected from the three plots were necromass traps were available (Pinto and Cuesta, 2019). At each of those plots, five subplots had necromass traps installed. From each necromass trap we took one sample, which consisted of five randomly selected leaves. In total we collected 15 necromass samples from three plots (Table 1).

Soil samples were collected at every plot. At each plot we selected the centre subplot and two randomly selected subplots from which to sample soils. From each subplot we took one sample, which consisted of 10 "pinches" of surface soil randomly taken across the subplot (we removed the necromass layer to access the surface soil where necessary). We unintentionally sampled one extra subplot at one of the plots (MALO_01, Table 1). In total we collected 43 soil samples from 14 plots (Table 1).

All samples were collected with gloved (latex) hands and wrapped in aluminium foil in the field, making sure no contact was made with the skin to avoid lipid contamination. Samples were bagged and placed in a cold storage (5 $_\circ$C) until further analysis at the University of Amsterdam.

## 2.2 *n*-Alkane extraction and quantification

Leaf data was previously extracted and quantified by Teunissen van Manen et al. (2019). Necromass and soil samples were analysed following the same protocol. Necromass and soil samples were freeze dried and milled to powder. Soil samples were analysed for carbon content in an Elemental Vario El Cube CNS analyzer. Necromass samples were assumed to contain c. 50% carbon. Depending on carbon content, between 0.1-0.5 g of sample was extracted using a Dionex 200 accelerated solvent extractor (ASE). With every extraction set we ran a blank sample that was treated as if it were a regular sample. The internal standard consisted of a mixture of 5α-androstane, androstanol, and erucic acid (0.33 µg/µL per compound) 40 µl of internal standard was added to each sample prior to extraction. The extracted solution was dried under a steady stream of $N_2$ to remove the solvent, and re-dissolved in 1mL hexane.

The *n*-alkane fraction was obtained by eluting a 10 mL solid phase column (approximately 1.5 g of silica gel, 5% deactivated $H_2O$, previously conditioned with acetone, dichloromethane and hexane) with approximately 7mL hexane in multiple steps. The resulting *n*-alkane fraction was again dried under a stream of $N_2$, re-dissolved into 1 mL hexane, and finally analysed using gas chromatography-mass spectrometry (GC-MS) with quadruplet detection in full scan mode. The GC-MS protocol was as follows: the sample was injected in a DB5 column (30 m) under constant flow of helium gas at 0.8 mL/min. Temperature programming was as follows: start at 50 °C (hold 2 min); first ramp at 60 °C/min to 80 °C (hold 2 min); second ramp at 20 °C/min to 130 °C (no hold); third ramp at 4 °C/min to 350 °C (hold 10 min).

Identification and quantification of $n$-alkanes was done by comparing measurements with a known mixture of $n$-alkanes in the range $C_{25}$-$C_{33}$, and the internal standard employing the Thermo Xcalibur® software. Limit of detection (LOD) was set at 3x the base level, any concentration below was set to "not found". The resulting $n$-alkane dataset was standardized to sample weight (grams of dry sample used for extraction). To estimate measurement variability, replicate measurements were done on four necromass and 26 soil samples. Fourteen replicate measurements of the leaf samples were previously measured and reported on in detail Teunissen van Manen et al. (2019).

Not all samples retrieved from the field yielded a robust $n$-alkane signal. In three samples we measured only sporadic $n$-alkane peaks (Table 1). In total, 86 leaf samples, 13 necromass samples, and 51 soil samples were considered robust data and were used for further data analysis (Table 1).

## 2.3 Data analysis

### 2.3.1 Standardization

The measured $n$-alkanes ranges differed between leaf, necromass and soil samples ($C_{23}$-$C_{33}$, $C_{21}$-$C_{33}$, and $C_{15}$-$C_{33}$, respectively). In order to compare the datasets, we standardized the measured $n$-alkane range to the shortest range, between $C_{23}$-$C_{33}$. We found this had no visible effect on the results, likely because the chain lengths below $C_{23}$ contained only a small proportion of the total $n$-alkane fraction, which has also been observed before (Ardenghi et al., 2017). The $n$-alkane concentrations of replicate samples were averaged before continuing data analysis.

### 2.3.2 Multivariate analysis of $n$-alkane patterns

We standardized the $n$-alkane data to the total $n$-alkane concentration (CONw) to obtain the relative abundance of each chain length (%). Where CONw is the sum of all $n$-alkane concentrations standardized to dry sample weight (ng/g of dry sample). We used these data as input for non-metric multidimensional scaling (nMDS) analysis on each sample type and on all sample types combined (four analyses in total). The aim was to identify the major $n$-alkane patterns defining each sample type and to also compare the sample types. We performed the nMDS using the metaMDS function, from the 'vegan' package (Oksanen et al., 2018) in RStudio (R Core Team, 2017), with Euclidean distance matrix and disabling the default data transformation intended for species community data. We fitted the environmental variables (temperature, humidity and precipitation) to the nMDS in order to help identify environmental correlations with the $n$-alkane patterns. We also used the standardized $n$-alkane data to produce the transect average $n$-alkane distributions (and standard deviation) per sample type.

### 2.3.3 *n*-Alkane metrics

We calculated three common metrics used in *n*-alkane biomarker studies, namely the average chain length (ACL), the normalized ratio between $C_{31}/C_{29}$, and the carbon preference index (CPI)(hereafter, 'the metrics'). The ACL was calculated following the definition by Bush & McInerney (2013) Eq. (1):

$$ACL_{23-33} = \frac{\Sigma(C_n \times n)}{\Sigma C_n}$$

(1)

Where $C_n$ is the concentration of *n*-alkane per gram of dried sample and n is the number of carbon atoms of an *n*-alkane between $C_{23}$ and $C_{33}$. The ratio between $C_{31}$ and $C_{29}$ *n*-alkanes was calculated following the Bush & McInerney (2013) Eq. (2):

$$ratio = \frac{C_{31}}{C_{31}+C_{29}}$$

(2)

The *n*-alkane CPI values were calculated following the definition of Marzi et al. (1993) Eq. (3):

$$CPI_{23-33} = \frac{[\Sigma_{odd}(C_{23-31}) + \Sigma_{odd}(C_{25-31})]}{2(\Sigma_{even}C_{24-32})}$$

(3)

Where $\Sigma_{odd}$ is the sum of all concentrations of odd chain *n*-alkanes between and including the indicated *n*-alkane chain length ranges, and $\Sigma_{even}$ is the sum of all concentrations of even chain *n*-alkanes between $C_{24}$-$C_{32}$.

### 2.3.4 Environmental correlations

We generated a correlation matrix with the environmental variables (altitude, temperature, humidity and precipitation), the metrics and the nMDS axes per sample type, to identify: (1) whether the *n*-alkane metrics captured dominant changes in the *n*-alkane patterns (the nMDS), and (2) whether the *n*-alkane signal (the metrics and the nMDS axes) and the environmental variables correlate. The Pearson's linear correlation matrices were calculated using the 'corrplot' (Wei and Simko, 2017) and 'Hmisc' (Harrell and Dupont, 2019) packages in RStudio (R Core Team, 2017). We adopted a significance threshold value of $p < 0.01$ because the bulk of our analysis are correlations; as is convention (Teunissen van Manen et al., 2019; Tipple and Pagani, 2013). All data analysis was done in RStudio using the base functions (R Core Team, 2017) and the 'tidyverse' packages functions (Wickham, 2017).

## 3 Results

The replicate measurements showed that absolute concentration of *n*-alkanes varied more than 10% from the mean (CV%) in the majority of the replicate samples, with CV% values ranging from 11.5% to 48% (Table A1). The high variability in absolute

concentration in the replicate samples could be due to sample heterogeneity despite the elaborate sample homogenization process prior to extraction. Despite the high variability in absolute concentrations, the relative distributions of the replicate samples showed almost no variability (Fig. B1), giving us confidence that the metrics and relative abundances presented are robust.

The transect average $n$-alkane patterns of each sample type reflected a typical higher terrestrial plant distribution, with a clear odd-over-even distribution between the ranges of $C_{23}$ and $C_{33}$ (Fig. 2, Fig. B1). The three sample types had similar average distributions of the chain lengths (Fig. 2). Specifically, they shared dominant $C_{29}$ and $C_{31}$ chain lengths, with an average contribution of $\pm$30-40% of CONw each (Fig. 2). Although the transect average proportion of $C_{27}$ differed somewhat per sample type, the variability across the transect is very high, in necromass in particular (Fig. 2).

### 3.1 Identification of the dominant changes in the $n$-alkane patterns

### 3.1.1 Description of the patterns

The nMDS analyses exposed the variability in the $n$-alkane patterns of each sample type (Fig. 3 a,b,c) and how the sample type patterns compare (Fig. 3d). Notably, the first axis of all the nMDS plots were driven by a shift in relative abundance from shorter $n$-alkanes ($\leq C_{29}$, lower end of the axis) to longer $n$-alkanes ($>C_{29}$, higher end of the axis) (Fig. 3a,b,c,d).
It is less clear what variability in the $n$-alkane patterns drove the samples to spread along the second nMDS axis. However, some even numbered chain lengths were placed on the higher end of the second axis of the leaf and soil nMDS plots (Fig. 3a,c). This indicates that the samples on the higher end of the second axis of the leaf and soil nMDS plots had relatively higher abundances of those even numbered chain lengths (Fig. 3a,c).

The nMDS analysis of all sample types together shows that the different sample types did not cluster or separate from each other, although the soil data did scatter less than the leaf and necromass data (Fig. 3d). The lack of separation and significant clustering indicates that the sample type patterns were not significantly different or distinguishable from each other.

### 3.1.2 Identification of the signal in terms of metrics

For all sample types, correlations between the first nMDS axis and the ACL and ratio metrics were significant (Fig. 4, Table 2), which indicates that the change in patterns captured by the first nMDS axis of all sample types are explained by shifts in ACL and ratio (Fig. 4). It should be noted that these two metrics were highly correlated across sample types (leaf: $r = 0.9$, p-value <0.001; necromass: $r = 0.8$, p-value <0.001; soil: $r = 0.9$, p-value <0.001) (Fig. 4, Table 2). The high correlation between ACL and ratio indicates that both metrics captured shifts in relative abundances of shorter $vs.$ longer $n$-alkane chain lengths, and accurately described the dominant shifts in the $n$-alkane patterns of all sample types in this study. Notably, the CPI metric

did not correlate with the first nMDS axis of any sample type (leaf: r = 0.1, p-value = 0.389; necromass: r = -0.2, p-value = 0.426; soil: r = -0.2, p-value = 0.291) (Fig. 4, Table 2).

Correlation between the second nMDS axis and the metrics differed per sample type. The second axis of the leaf data nMDS correlated with the ACL and CPI metrics (r = -0.4, p-value < 0.001 and r = -0.3, p-value = 0.005, respectively) (Fig. 4a, Table 2). However, these correlations were non-significant when the samples separating from the main clusters on the top left of the leaf nMDS plot were removed from the analysis (Fig. 3a, Fig. C1). These four samples represent a single species sampled at a

225 one plot at high altitude (Teunissen van Manen et al., 2019). Therefore, the changes in the leaf *n*-alkane pattern captured by the second nMDS axis could not be identified by commonly used metrics.

The second axis of the necromass data nMDS did not significantly correlate with any of the metrics (Fig. 4b, Table 2). The changes in the necromass *n*-alkane pattern captured by the second nMDS axis could not be identified by commonly used

metrics.

The second axis of the soil data nMDS correlated with the CPI metric (r = -0.5, p-value < 0.001), which indicates that the change in patterns captured by the second nMDS axis of the soil data was partially reflected by shifts in CPI (Fig 4c, Table 2). However, the correlation was not complete, therefore some changes in the soil *n*-alkane patterns (as captured by the second

nMDS axis) remained un-attributable to commonly used metrics.

### 3.2 Environmental signal of the *n*-alkane patterns

The environmental variables (altitude, temperature, humidity and precipitation) highly co-correlated (Fig 4a,b,c, Table 2), in particular altitude and temperature were exact opposites (r = -1, p-value <0.00, regardless of sample type). This was to be expected as we sampled along a single altitudinal gradient. In order to avoid repetition, we will focus the remainder of the

240 results and the discussion on the temperature, humidity and precipitation variables only.

The dominant changes in the leaf *n*-alkane patterns, identified as changes in the ACL and ratio metrics, correlated positively with the environmental variables (Fig. 3a, Fig. 4a, Fig. 5, Table 2). Although CPI did not significantly correlate with any of the pattern shifts captured by the nMDS (after removing the one species separating from the rest) (Fig. 3a), the leaf data CPI

did significantly correlate with humidity (r = -0.4, p-value <0.001) (Fig. 4a, Fig. 5, Table 2).

The dominant changes in the necromass *n*-alkane patterns, identified as changes in the ACL and ratio metrics, did not show any significant correlations with the environmental variables (Fig. 3b, Fig. 4b, Fig. 5, Table 2). This was most likely due to the limited number of samples that were collected along the transect (Table 1).

The dominant changes in the soil $n$-alkane patterns, identified as changes in ACL, ratio and CPI, correlated with temperature in particular (r = 0.6, p-value <0.001, r = 0.5, p-value <0.001 and r = -0.4, p-value <0.001, respectively) (Fig. 3c, Fig. 4b, Fig. 5, Table 2). Additionally, soil ACL and CPI also correlated with precipitation (r = 0.4, p-value = 0.001 and r = -0.4, p-value = 0.005) (Fig. 4c, Fig. 5, Table 2).

## 4 Discussion

### 4.1 $n$-Alkane patterns and what they signal at different stages of source degradation

#### 4.1.1 Do the $n$-alkane patterns degrade significantly?

The $n$-alkane patterns (i.e. the relative chain length distribution) of all sample types reflected those typical of higher terrestrial plants (Eglinton and Hamilton, 1967), suggesting that although absolute $n$-alkane concentrations may change due to degradation, the $n$-alkane chain length distribution patterns do not degrade considerably (Fig. 2). This is corroborated by the nMDS results, which indicate the $n$-alkane patterns from all stages of degradation are indistinguishable from each other, although soil $n$-alkane patterns did show less variance than necromass and leaves $n$-alkane patterns (Fig. 3d). These results fall in line with the results presented in the few studies that have compared leaf and soil $n$-alkanes, who also find similar distributions of leaf $n$-alkanes and soil $n$-alkanes (Howard et al., 2018; Tipple and Pagani, 2013). Large variance in leaf $n$-alkane patterns has been observed before (Bush and McInerney, 2013; Carr et al., 2014), but was never directly compared to necromass or soil $n$-alkane variability. Our results give a first indication that necromass $n$-alkanes patterns can vary to the same large extent that leaf $n$-alkane patterns do, whereas soil $n$-alkanes patterns exhibit a more homogeneous signal, most likely owing to the spatiotemporal mixing of the input (Howard et al., 2018; Wu et al., 2019).

An important confounding factor to consider with respect to the soil $n$-alkane signal is the potential input of root derived $n$-alkanes. Previous work in the Ecuadorian Andes has shown that roots of local plants can produce significant amounts of higher chain-length $n$-alkanes (Jansen et al., 2006). Several other studies have also reported a significant production of higher chain-length $n$-alkanes in plant roots, as reviewed by Jansen and Wiesenberg (2017). However, the absolute concentrations of higher chain-length $n$-alkanes produced by plant roots are usually an order of magnitude lower that those produced by plant leaves (Schäfer et al., 2016; Jansen and Wiesenberg, 2017). In addition, generally the $n$-alkane chain-length distribution patterns produced by the roots of a plant species vary substantially from the $n$-alkane patterns produced by the leaves of the same species to the point that the variability is as large as between the $n$-alkane chain-length distribution patterns produced by the leaves of two unrelated plant species (Jansen and Wiesenberg, 2017). Both observations also hold true for the previously mentioned study in the Ecuadorian Andes (Jansen et al., 2006). Jansen and Wiesenberg (2017) therefore concluded that the extent to which root derived $n$-alkanes confounds the observed relationship between leaf wax alkane patterns and those observed in the soil is strongly ecosystem dependent. Given that root and leave $n$-alkane chain-length distribution patterns of

the same species can be expected to vary substantially, in the present study the observed similarity between the *n*-alkane patterns in soils as compared to the leaves (Fig.2) suggests that the input of root-derived *n*-alkanes has not been an overriding factor in determining the *n*-alkane patterns in the soil here. Nevertheless, we cannot rule out that a minor part of the observed

differences between the *n*-alkane patterns in leaves and soils, including the mentioned relative muting of the soil *n*-alkane signal, was caused by input of root-derived *n*-alkanes.

Overall, our findings suggest that the *n*-alkane pattern does not degrade considerably as the plant source degrades along the transfer from leave via necromass to soil, but that some information is lost once reworked into soils and possibly mixed with

290 root derived *n*-alkanes.

### 4.1.2 Does the signal reflected in the *n*-alkane patterns degrade?

Our results indicate that the *n*-alkane patterns extracted predominantly vary in the relative abundance of longer *vs.* shorter *n*-alkanes, regardless of stage of degradation (Fig. 3a,b,c,d), suggesting that the *n*-alkane signal does not drastically degrade as the plant source material degrades. In our study, the ACL and ratio metrics reflected the *n*-alkane signal, independent of sample

type.

The CPI is a metric specifically developed to capture changes in odd-over-even predominance, and a decrease in CPI is generally accepted to reflect degradation (e.g. Marzi et al., 1993). Our results show that only the soil *n*-alkane signal reflects changes in CPI (Figs 3 and 4) which suggests that the processes leading to changes in CPI are related to soil processes.

Although the precise mechanisms are still subject to debate, a likely explanation is the de-novo genesis of *n*-alkanes without an odd-over-even predominance as a result of microbial alteration (Brittingham et al., 2017; Jansen and Wiesenberg, 2017; Rao et al., 2009; Wu et al., 2019). A decrease in CPI as a result of the input of root-derived *n*-alkanes in soils is unlikely because, as explained previously, these appear to have played a minor role in our study, and moreover the odd-over-even chain-length predominance of root wax *n*-alkane patterns is generally similar to that of leave derived *n*-alkanes (Jansen &

Wiesenberg, 2017). An important observation is that the CPI does not correlate with the ACL and ratio metrics in the present study (Fig. 4c), indicating that the dominant signal (changes in longer *vs.* shorter *n*-alkanes) is not affected by the secondary signal (changes in odd *vs.* even *n*-alkanes) in soils. This confirms previous indications that soil degradation processes do not drastically alter the ACL, but instead alter the odd-over even predominance (i.e. CPI) (Howard et al., 2018).

### 4.2 Environmental information in *n*-alkane signals and its preservation across sample types

We found that the systematic variance in longer *vs.* shorter *n*-alkanes in leaf and soil samples, also reflected in the ACL and ratio metrics, positively correlates with precipitation and, in particular, temperature (Fig. 4a,c, Fig. 5). Our results also show that the strength of the observed correlations diminishes slightly as the source material degrades (Fig. 4a,c, Fig. 5). This is also reflected in the lower variance observed in the soil ACL and ratio metrics (nMDS axis 1, Fig. 3d)..

Several plant physiological studies have shown that both the absolute amounts of leaf waxes produced as well as the relative chain-length abundance of their constituent *n*-alkanes may vary systematically with temperature and, to a lesser extent, water availability (e.g. Maffei et al., 1993; Zhang et al., 2004; Shepherd and Griffiths, 2006). While we did not explicitly measure it, $pCO_2$ is another factor that can be expected to vary along the altitudinal transect. However, several studies have shown that while a $pCO_2$ gradient may influence the overall vegetation composition along the gradient, in particular if it leads to a shift from C3 to C4 vegetation, the influence of a change in $pCO_2$ on leaf wax alkane patterns within a plant species is very limited (Huang et al., 1999; Wiesenberg et al., 2008). Since we sampled the same plant genera along our transect, which all consisted of C3 plants, it is therefore unlikely that the systematic variation in *n*-alkane patterns with altitude found were caused by the $pCO_2$ gradient. Therefore, the observed correlations in our study might reflect changes in environmental conditions, particularly in temperature, along the altitudinal gradient studied; with the expression of the signal being the strongest in the leaf material (Fig. 4a,c, Fig. 5).

Others have also observed a systematic relationship between leaf and soil *n*-alkane patterns as expressed in the ACL values along a (latitudinal) temperature gradient (Bush and McInerney, 2015; Tipple and Pagani, 2013). Zooming in on the preservation of the signal, the results were mixed.. Bush and McInerney (2015) found a stronger correlation between soil ACL and temperature than plant ACL and temperature in the USA. Also in the USA, Tipple and Pagani (2013) found that the soil and plant ACL correlation with temperature were in the same order of magnitude. However, the difference between our results and previous work can be explained by the differences in the leaf data used in these studies, where one study focusses on two particular species and the other study sampled both woody and herbaceous plants (Bush and McInerney, 2015; Tipple and Pagani, 2013). In contrast, a third study reporting on leaf and soil ACL data from South Africa, found no relation to temperature or precipitation but they noted this was due to the short environmental gradient sampled (Carr et al., 2014). The muted correlation of ACL with temperature in soil *n*-alkanes in our study as compared to the leaf *n*-alkanes a is most likely the result of soil *n*-alkanes reflecting an average of a larger spatial/temporal scale than that reflected in leaves (Howard et al., 2018; Wu et al., 2019). The overall reduction of the range of variability in the soil *n*-alkane patterns (compared to leaf *n*-alkane patterns) supports this idea (Fig. 3d), but further research is needed. It is unlikely that the muted correlation is caused (preferential) degradation of the *n*-alkane signal, as ACL and CPI are uncorrelated (Fig. 4c).

A systematic temperature and/or precipitation signal recorded in leaf wax *n*-alkanes and subsequently preserved in the soil, as our results seem to indicate, would offer an exciting addition to our palaeoecological proxy toolbox. However, the relationships that were observed must be interpreted with caution. While the correlations are statistically significant, the signal shows an appreciable degree of noise (Figs. 3 and 4), indicative of the presence of other factors in addition to the environmental parameters that influence the *n*-alkane patterns. Indeed, in addition to environmental factors, also genetic and ontogenetic factors are significant determinants of *n*-alkane patterns in leaf waxes, and the relative importance of each of these is most

likely strongly ecosystem dependent (Jansen & Wiesenberg, 2017). Moreover, correlation does not prove causation, and while unlikely in the light of the available literature that was previously discussed, we cannot completely rule out that the observed correlation of ACL with temperature and precipitation is the result of a hidden parameter that was not considered, or even pure coincidence.

A second environmental correlation to consider is the observed inverse correlation of the CPI metrics of soil *n*-alkanes with temperature and precipitation (Fig. 4c). A similar inverse correlation has been observed before by others in soils and has been attributed to increased microbial degradation of *n*-alkanes under favourable (warm and wet) conditions, independent of local standing vegetation (Luo et al., 2012; Rao et al., 2009). Along an altitudinal transect in Peru similar to our study, lower CPI values at lower elevations (warmer) have also been observed in soils and suspended river sediments (Feakins et al., 2018; Wu et al., 2019). Contrastingly, Bush and McInerney (2015) did not find a clear correlation between soil CPI and environment. However, the Bush and McInerney (2015) study was done along a latitudinal gradient in the USA so it is possible that the relationship is not perceived in all ecosystems, and instead particular to the tropical settings of this study. Again, this is testimony to the ecosystem dependency of the dominant factors driving *n*-alkane patterns in vegetation and soils (Jansen and Wiesenberg, 2017).

### 4.3 Implications for *n*-alkanes as a proxy for past environmental change

### 4.3.1 Reconstruction of vegetation composition

Together with previous findings (e.g. Bush and McInerney, 2015; Teunissen van Manen et al., 2019; Tipple and Pagani, 2013), our results suggest that *n*-alkane patterns vary predominantly in the relative abundances of longer *vs.* shorter *n*-alkanes, which are consistently reflected in the ACL and ratio metrics in our study, regardless of degradation level. This indicates that, under the specific environmental circumstances in our study, the information with respect to relative abundance of *n*-alkanes of various chain-lengths is preserved throughout their journey from the plant leaf to the soil. This is important as it forms the basis of palaeoecological vegetation reconstructions based on the difference in relative *n*-alkane abundances indicative of different (groups of) plant species (e.g. Jansen et al, 2013). Our results support the mounting evidence that in many ecosystems, including the one represented in our study, interpreting the *n*-alkane biomarker record based on knowledge from modern *n*-alkane patterns in principle is valid. However, the large differences in variability between the sample types should be noted, leaf *n*-alkanes being highly variable and soil *n*-alkanes less. The decreased variability of soil *n*-alkane patterns has been noted before and has been attributed to degradation of the *n*-alkane signal in soils and the averaging qualities of soil samples (Bush and McInerney, 2015; Howard et al., 2018; Wu et al., 2019). Therefore, when comparing ancient *n*-alkane records to modern leaf data the larger variability of leaf *n*-alkane patterns and responses should be kept in mind, as sedimentary *n*-alkane records are unlikely to reflect the same range of responses. Additionally, results from Wu et al (2019) suggest that soil *n*-alkanes are a

380 quantitatively important source of sedimentary *n*-alkane records. Therefore, although our results suggest that the *n*-alkane signal remains similar, basing *n*-alkane record interpretations of modern leaf data could lead to over- or underestimation of the reconstructed change in vegetation composition, and shifts in environmental composition inferred from it. One way to overcome this, is to interpret the ancient *n*-alkane record based on calibrations of modern soil *n*-alkanes rather than on modern plant *n*-alkanes. Of course, one should keep in mind that in our present study we only considered soil samples from the very

top of the soil profile. In order to make a palaeoecological reconstruction over time, the *n*-alkane patterns must subsequently also be preserved over time in the soil and be present in a chronological stratification. Whether or not this is the case will again be strongly ecosystem dependent (Jansen & Wiesenberg, 2017).

### 4.3.2 Reconstruction of temperature and/or precipitation

As indicated, the observed correlations of the ACL and ratio metrics with temperature and to a lesser extent precipitation may

constitute a first step to exploring the possibility of using such preserved leaf wax *n*-alkane patterns as temperature and/or precipitation proxy. With respect to preservation of the signal, it was encouraging to see that, although the strength of the correlations between the metrics (ACL and ratio) and the environment decreased as the source material degraded (Fig. 4, Fig. 5), the *n*-alkane patterns were muted but not obliterated. Nevertheless, as a result of the degradation and smoothing of the signal once in the soil, the changes in the *n*-alkane biomarker signal likely underestimate past environmental change and it is

doubtful that a quantitative reconstruction of the magnitude of change can be based on the *n*-alkane record alone. Corrections to counter the ancient *n*-alkane signal potential underestimation of inferred changes have been proposed, specifically using CPI as an indicator of *n*-alkane signal degradation (Buggle et al., 2010), and may serve as a starting point of further study. However, in addition to potential degradation hindering a quantitative interpretation, the more fundamental point of the observed noise within the correlations and the co-correlation of temperature and precipitation must be considered. Together

with the variable results obtained by the few other studies that considered systematic temperature and/or precipitation dependency of the *n*-alkane signal in the context of an environmental reconstruction, this indicates that there still is much research needed to rigorously test and further develop such a proxy. In particular, the fact that the leaf wax *n*-alkane signal represents an ecosystem-specific interplay of environment, genetics and ontogeny poses a challenge. This means that the proxy will not be universally applicable, but only in situations where the temperature and/or precipitation signal has a significant

influence on the *n*-alkane patterns with respect to the other factors that govern it. Even then, those other factors must be filtered out as much as possible, especially if we are looking for a quantitative application rather than a qualitative indication of a direction of change. Possible steps in such an endeavour may be to combine a possible application as temperature and/or precipitation proxy with existing modelling approaches aimed at distilling vegetation composition information (i.e. the genetic component) out of the mixed *n*-alkane signal in soils, such as with the VERHIB model (Jansen et al., 2013). In addition, a

combined application with compound specific δD analysis of *n*-alkanes of the dominant chain lengths in the signal may help unravel a potential precipitation signal (e.g. Lane et al., 2018). Our study should be seen as an encouraging incentive for such future research, with the ultimate goal not of finding a silver bullet proxy for temperature and/or precipitation changes, but

rather an additional proxy that when applied in a multi-proxy approach with other palaeoecological proxies can help us reconstruct past environments with increasing accuracy and completeness.

The value of CPI as a proxy has been up for discussion, as plant data varied largely and with no systematic variability with environmental conditions observable (Bush and McInerney, 2013; Carr et al., 2014; Feakins et al., 2016; Wang et al., 2018). However, this illustrates how knowledge from leaf *n*-alkane signals does not directly translate to the *n*-alkane biomarker record, as it overlooks soil processes which can imprint environmental information in the *n*-alkane signal (this study, Wu et al., 2019).

In our study and in previous work, soil CPI has been observed to vary with temperature (Luo et al., 2012; Rao et al., 2009) suggesting that the *n*-alkane biomarker signal has a secondary way of reflecting environmental conditions due to soil processes, most likely microbial degradation of the *n*-alkane signal by altering the even-chain length in the *n*-alkane pattern. However, this signal is likely dependent of the study region, as studies conducted outside of tropical settings have not observed the correlation between soil CPI and environmental (warm/wet) conditions. Therefore, while our study confirms the general view

that CPI is a valuable indicator of degradation, it seems unlikely that other environmental information can be distilled from it.

**5 Conclusion**

Our study of the *n*-alkane patterns in leaf waxes of the same plant genera sampled together with leaf necromass and the uppermost soil layer along a tropical altitudinal transect offered a unique opportunity to study the preservation of the *n*-alkane patterns and specifically any potential environmental information contained therein along the leaf-necromass-soil chain. In

addition, it gave us the opportunity to explore a possible, tentative relationship between *n*-alkane patterns and temperature and precipitation. Our results showed that degradation of *n*-alkane patterns did occur, as indicated by a muted signal and an increase in CPI in the soil samples. However, the similarity between the overall *n*-alkane patterns as well as the ACL in leaf and soil samples, showed that such degradation did not obliterate the *n*-alkane patterns. More importantly, the fact that the ACL and ratio metrics significantly correlated with the temperature and precipitation gradient along the transect in all sample types

(leaves, necromass and soil) shows that the environmental information stored in the *n*-alkane patterns was preserved along the journey of the *n*-alkanes from leaf to soil. Further interpretation of that environmental signal seems to hint at a direct relationship between temperature and precipitation, and the *n*-alkane patterns that might be explored as an additional palaeoecological interpretation of leaf wax derived *n*-alkane patterns stored in soils and sediments, next to the more established interpretation as proxy for shifts in vegetation composition. Results from the plant physiological literature as well as a limited

number of other studies that explored such a relationship in the field, justifies the conclusion that this relationship warrants further exploration. However, one should keep in mind that the present study was not explicitly designed to establish a causal relationship between *n*-alkane patterns and environmental factors. For this process-oriented plant physiological studies under controlled conditions are recommended. Moreover, even if a causal relationship is rigorously established, many other

confounding factors may limit its application as palaeoecological proxy. A causal relationship between *n*-alkane patterns and

445 temperature/precipitation may be obscured by other co-varying factors such as a shift in vegetation composition over time. In addition, while we found good preservation of the information contained in leaf-wax *n*-alkane patterns in the youngest soil layer, subsequently such a signal must be preserved in chronological order in soils or sediments if it is to be used to track the relationship over time. Whether or not such preservation occurs will be highly ecosystem dependent. Therefore, we explicitly do not see the observed relationship between *n*-alkane patterns and temperature/precipitation as a silver bullet new proxy.

Rather we see it as an interesting addition to the existing and ever-expanding suite of palaeoecological proxies that warrants further investigation to establish to what extent and in which specific ecosystem-dependent settings it might help improve the accuracy and precision of multi-proxy palaeoecological reconstructions.

**Data availability**

All data presented in this manuscript is available on Figshare: https://doi.org/10.21942/uva.10299068

## Appendices

### Appendix A

**Table A1: Sample replicates table, denoting: the replicate sample field code (sample), replicate sample type (type), the replicate sample average *n*-alkane concentration in ng/g of dried sample (CONw), the associated standard deviation (SD), coefficient of variance (CV(%)) and the number of replications done (N).**

| SAMPLE | TYPE | CONw | ±SD | CV(%) | N |
|---|---|---|---|---|---|
| 1 | necromass | 269.34 | 62.24 | 23.1 | 3 |
| 12 | necromass | 342.74 | 40.04 | 11.7 | 3 |
| 14 | necromass | 788.67 | 244.28 | 31.0 | 3 |
| 7 | necromass | 444.65 | 39.16 | 8.8 | 3 |
| BIOM 003 | soil | 227.71 | 32.84 | 14.4 | 2 |
| BIOM 006 | soil | 138.49 | 7.38 | 5.3 | 3 |
| BIOM 007 | soil | 167.98 | 28.26 | 16.8 | 6 |
| BIOM 010 | soil | 322.17 | 5.83 | 1.8 | 3 |
| BIOM 015 | soil | 251.93 | 65.95 | 26.2 | 2 |
| BIOM 018 | soil | 48.64 | 12.66 | 26.0 | 6 |
| BIOM 019 | soil | 37.01 | 1.55 | 4.2 | 2 |
| BIOM 022 | soil | 90.00 | 13.01 | 14.5 | 6 |
| BIOM 024 | soil | 127.41 | 2.69 | 2.1 | 2 |
| BIOM 025 | soil | 66.42 | 16.21 | 24.4 | 2 |
| BIOM 026 | soil | 136.65 | 24.95 | 18.3 | 3 |
| BIOM 027 | soil | 114.12 | 2.19 | 1.9 | 2 |
| BIOM 028 | soil | 237.27 | 32.63 | 13.8 | 3 |
| BIOM 032 | soil | 129.86 | 5.60 | 4.3 | 3 |
| BIOM 033 | soil | 166.04 | 30.86 | 18.6 | 2 |
| BIOM 036 | soil | 196.50 | 94.24 | 48.0 | 2 |
| BIOM 038 | soil | 57.83 | 13.13 | 22.7 | 3 |
| BIOM 040 | soil | 1094.09 | 155.69 | 14.2 | 3 |
| BIOM 046 | soil | 167.32 | 30.76 | 18.4 | 3 |
| BIOM 049 | soil | 396.95 | 15.78 | 4.0 | 5 |
| BIOM 050 | soil | 104.22 | 7.69 | 7.4 | 2 |
| BIOM 051 | soil | 136.15 | 42.57 | 31.3 | 3 |
| BIOM 053 | soil | 61.85 | 7.12 | 11.5 | 2 |
| BIOM 056 | soil | 158.34 | 25.50 | 16.1 | 3 |
| BIOM 057 | soil | 180.53 | 16.65 | 9.2 | 3 |
| BIOM 058 | soil | 136.85 | 34.39 | 25.1 | 2 |

**Figure B1: Sample fingerprints, relative abundances (%) of the *n*-alkane chain lengths per sample. Samples that were measured**
**repeatedly show average relative abundance (%) and standard deviation from the average relative abundance (lines), also see Table A1. Graph titles reference to sample field codes and sample type, where numbers without prefix indicate necromass samples, the BIOM prefix indicates soil samples and the GK, MCL, MTH, MBR, MCO and MOC prefix indicate plant samples (1/6).**

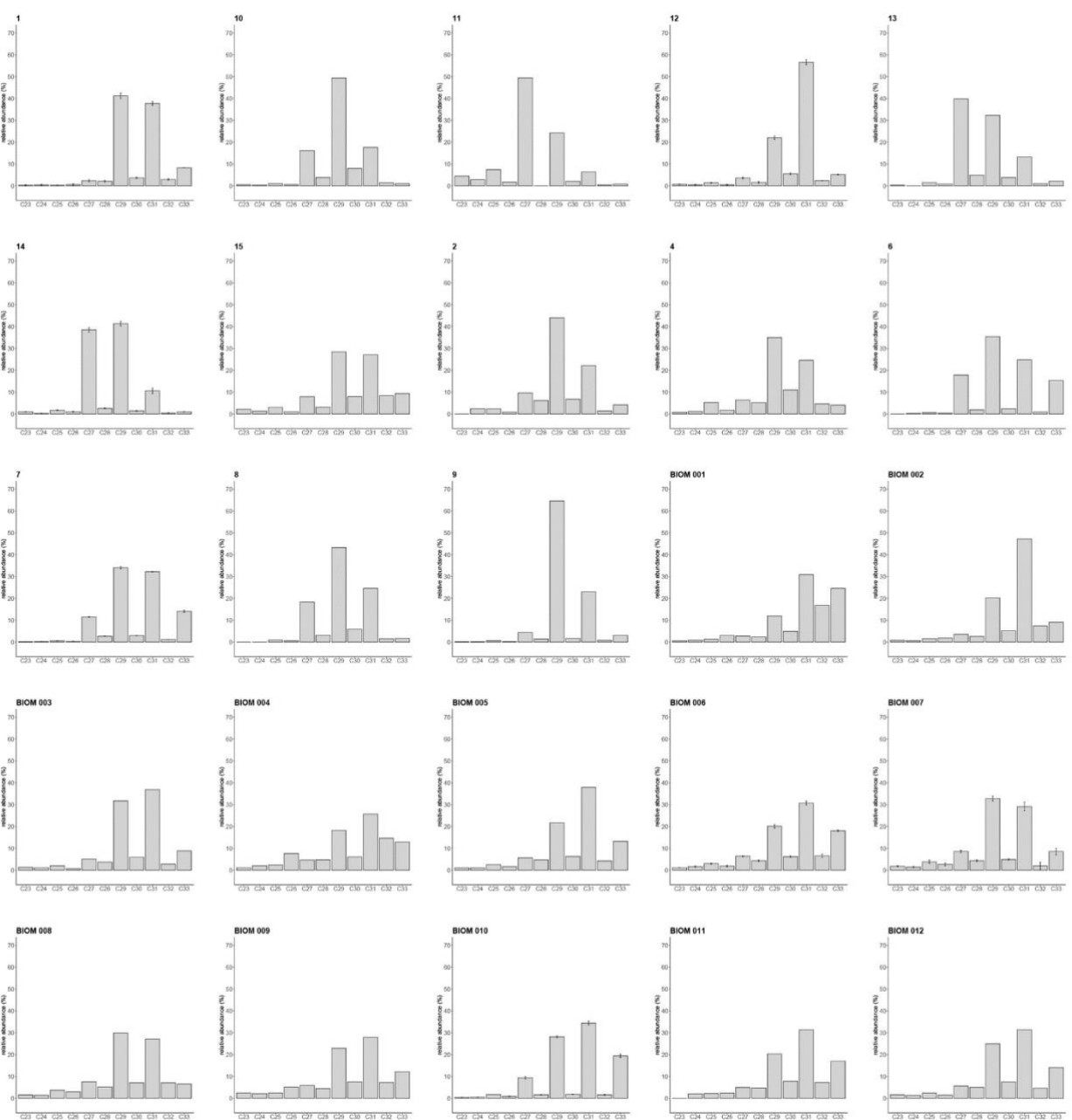

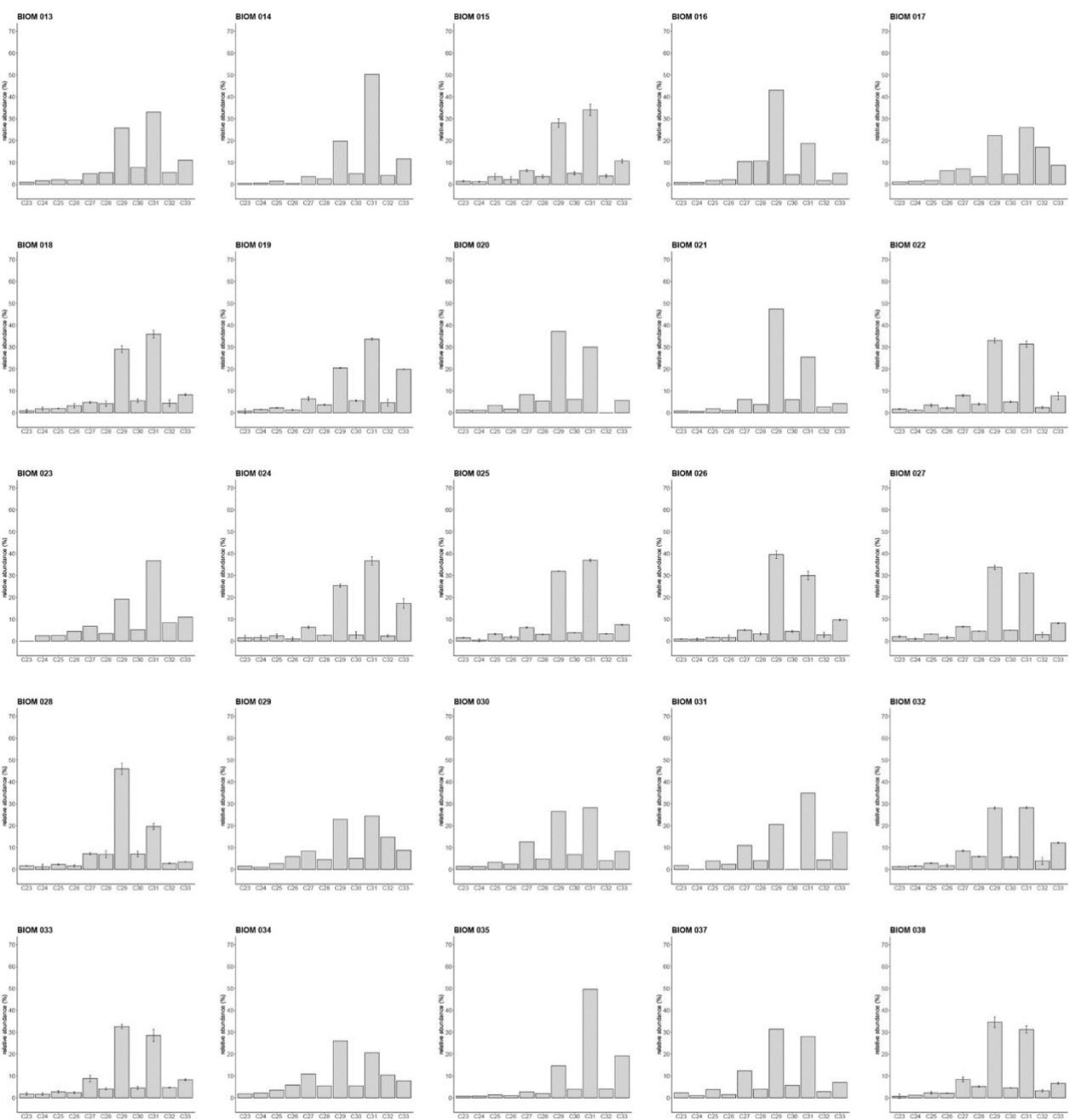

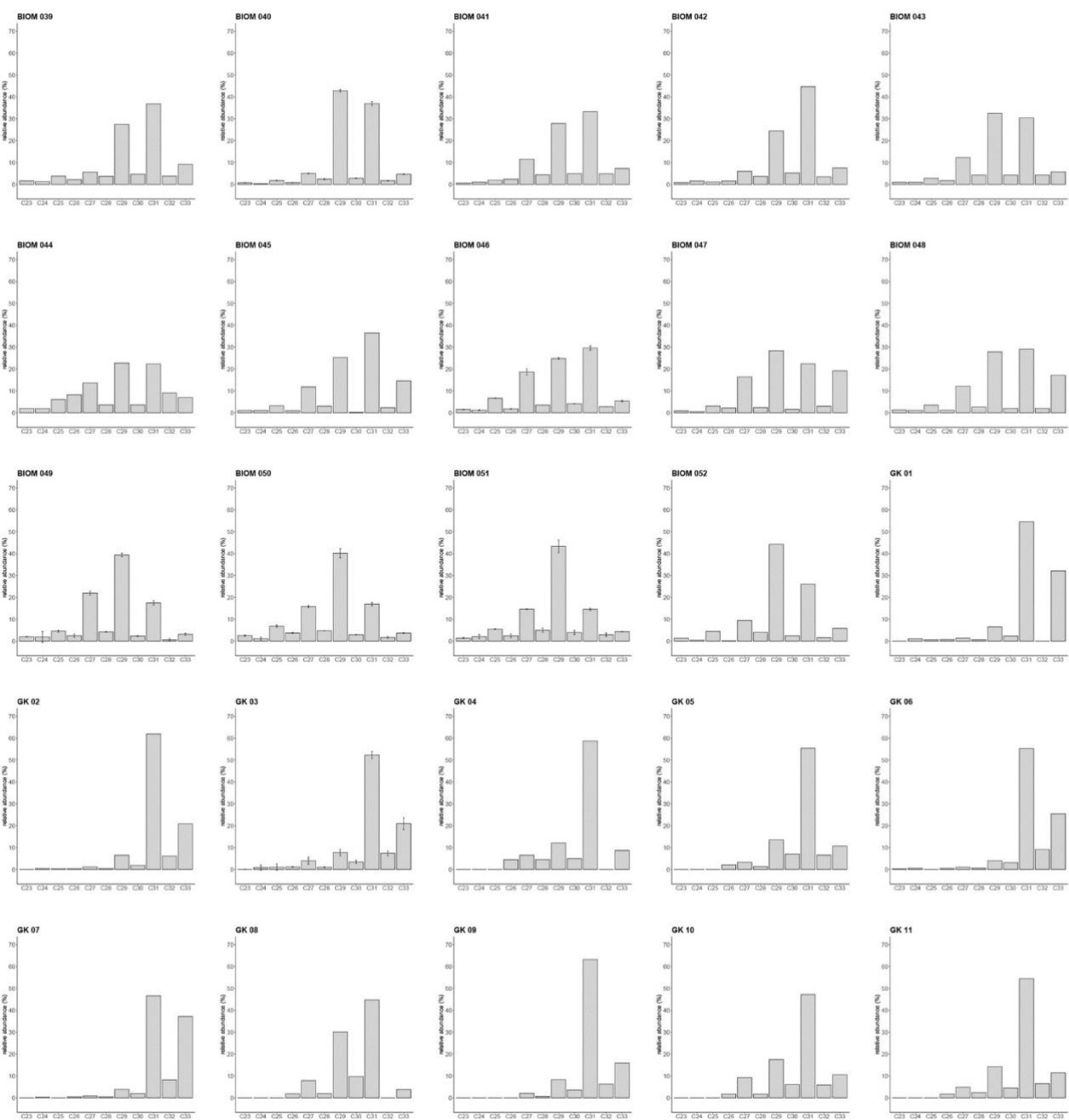

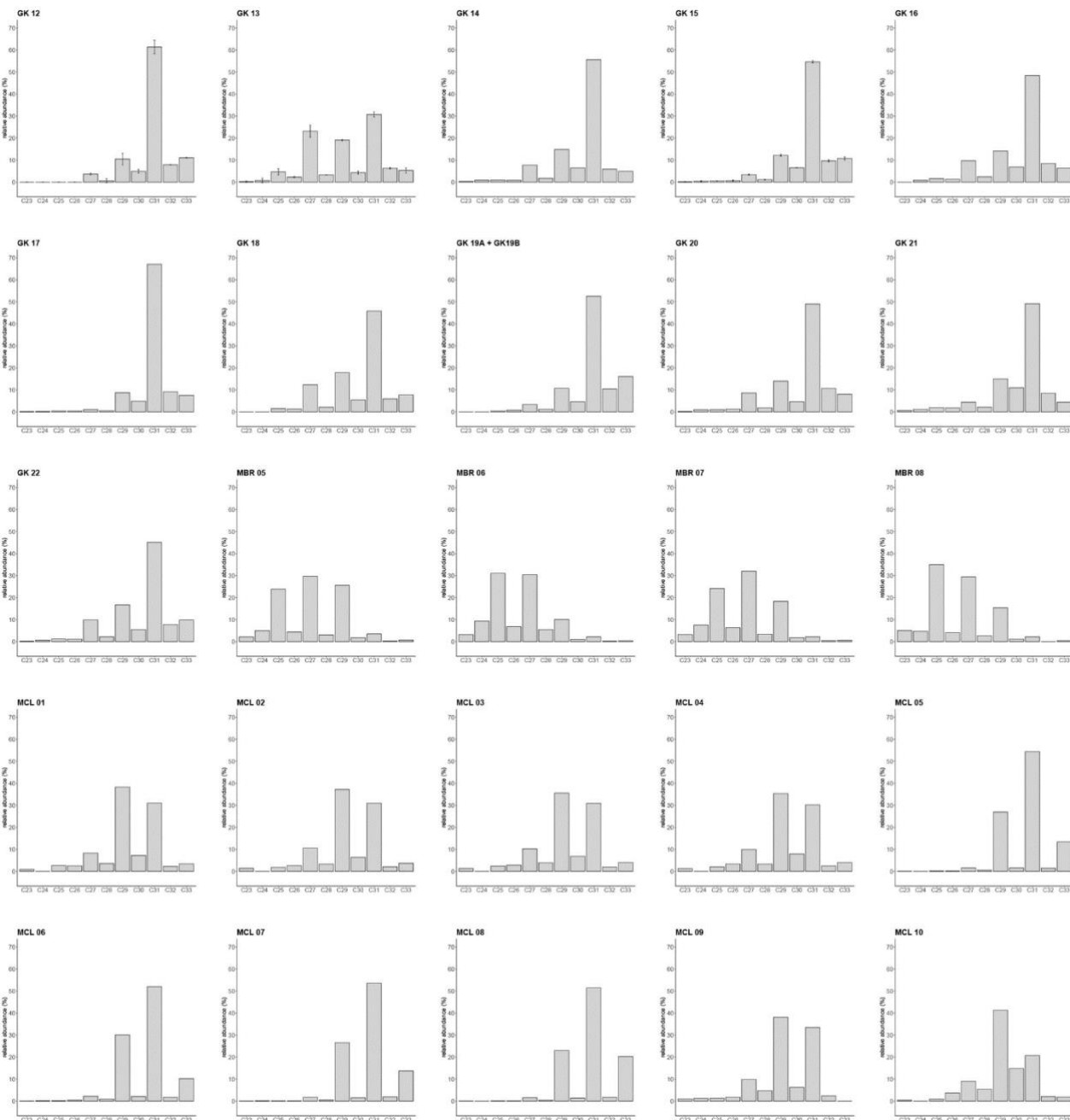

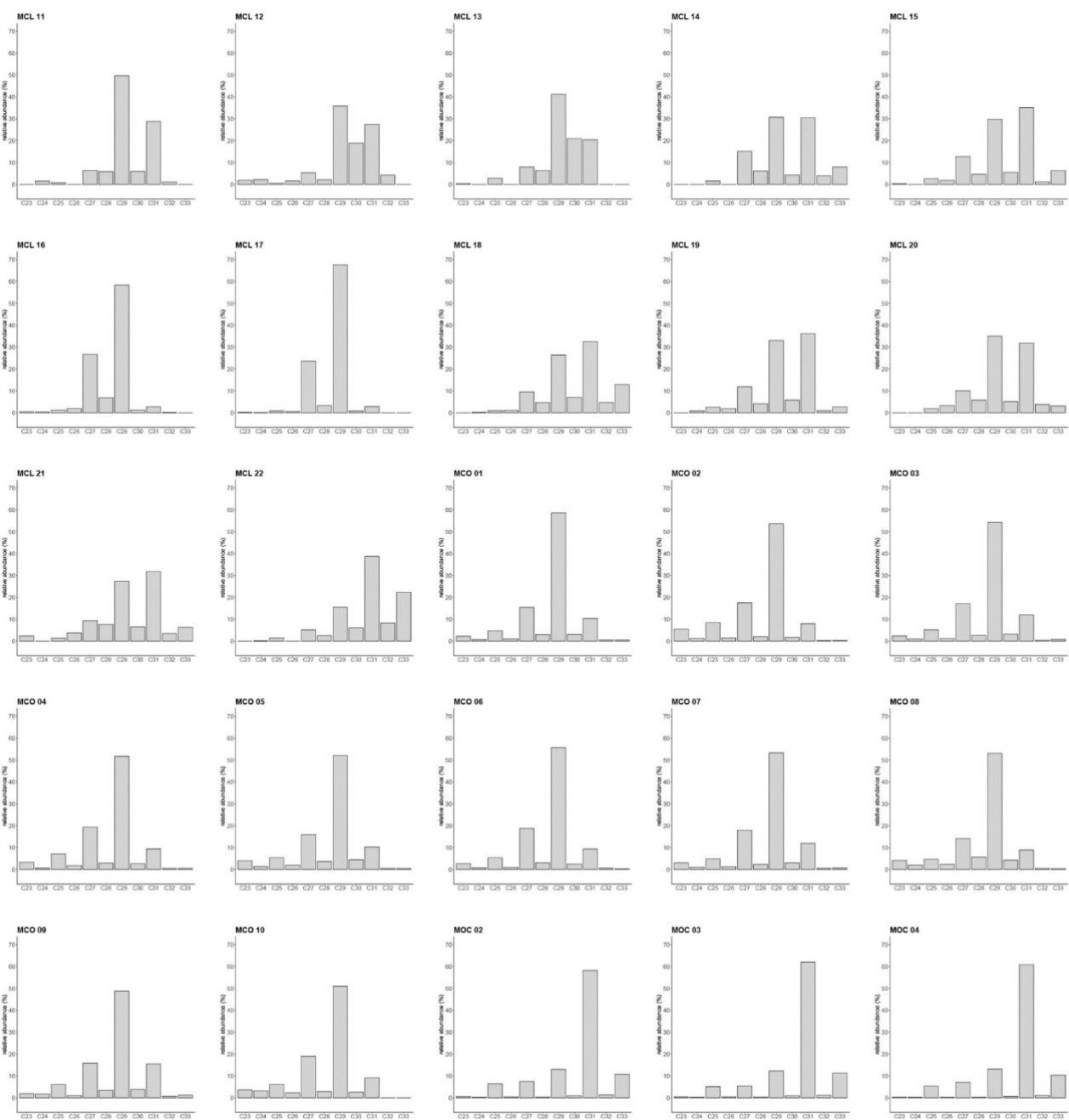

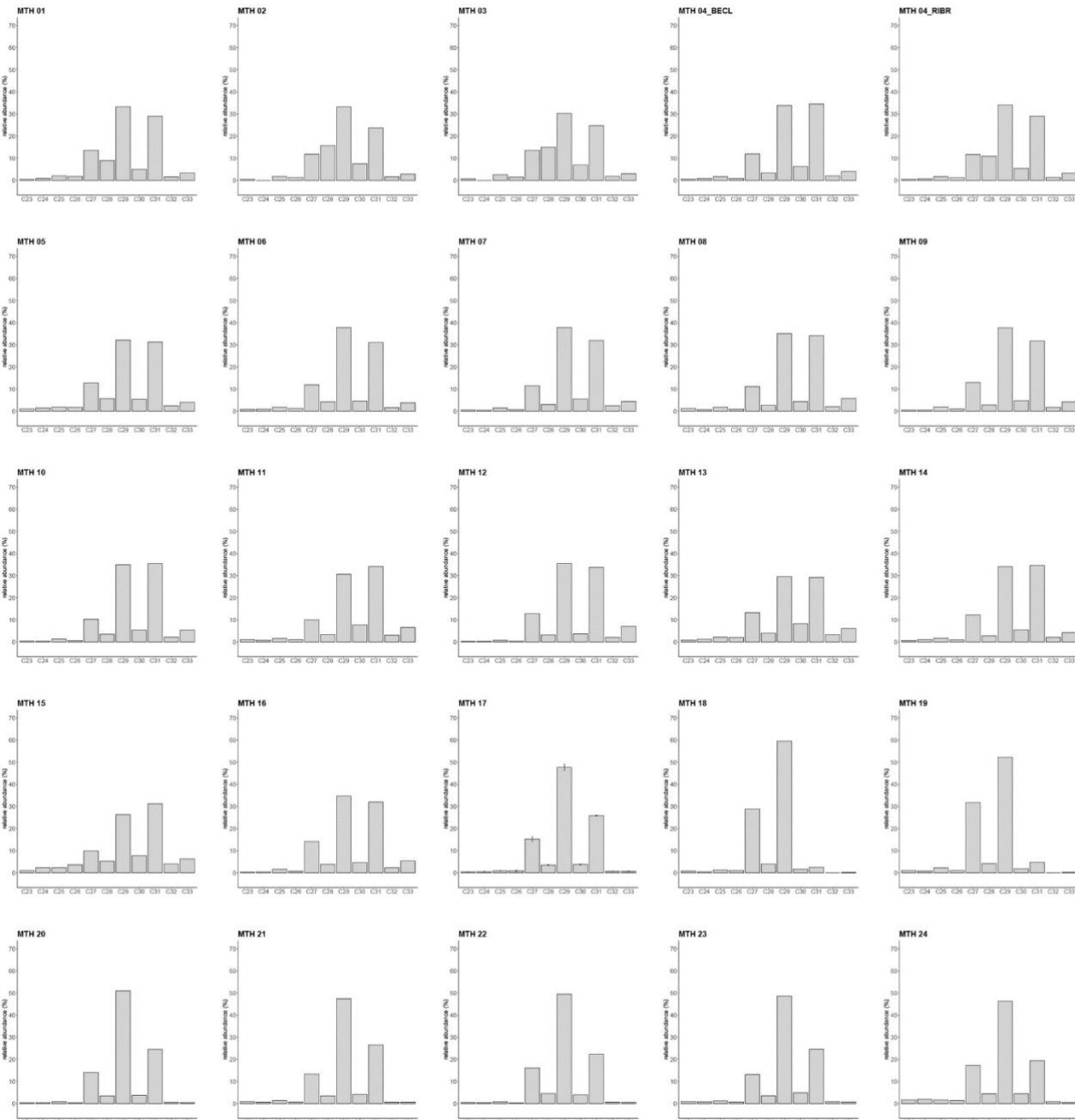

**Appendix C**

**Figure C1: nMDS (a, b) and correlation results (c, d) of the leaf samples without the *Miconia bracteolata* species samples (4 samples in total).**

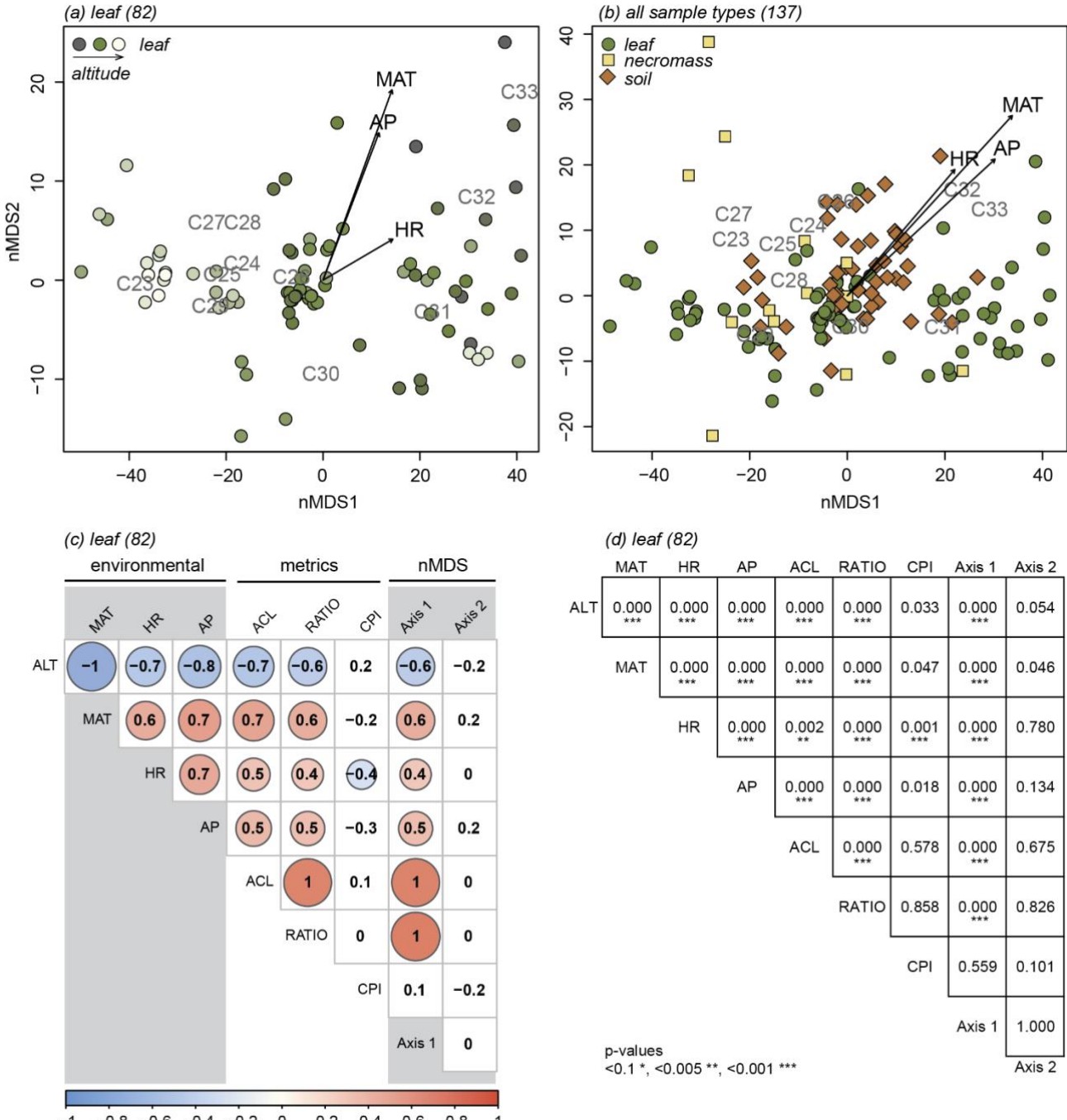

**Author contribution**

MTVM, WDG and BJ conceived and designed the study. MTVM, FCC and SLY facilitated and conducted fieldwork. MTVM did the lab analysis and analysed the data. MTVM, WDG and BJ wrote the manuscript. All authors contributed to the drafts of the manuscript and its final approval.

**Funding**

Financial support came the Institute for Biodiversity and Ecosystem Dynamics (University of Amsterdam). Additional funding came from the EcoAndes Project conducted by Consortium for the Sustainable Development of the Andean Ecoregion (CONDESAN) and United Nations Environment Programme (UNEP), funded by the Global Environmental Fund (GEF), cooperation agreement N° 4750.

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

**Table 1: Sampling table. Plot numbers refer to numbers in Figure 1. Plot codes refer to specific plots within the natural reserve, situated at their respective longitude and latitude. Altitude, temperature (MAT), humidity (HR) and precipitation (AP) data corresponding to each plot is included. Table notes what sample types were taken (sample type), how many were taken (sampled) and which samples were measured successfully and included in analysis (n).**

| plot # | plot code | reserve | latitude | longitude | altitude | MAT | HR | AP | sample type | sampled | n |
|---|---|---|---|---|---|---|---|---|---|---|---|
| 1 | MAPI_02 | Mashpishungo/Pambiliño | 0.1882 | -78.9128 | 632 | 21.6 | 99.8 | 2111 | soil | 3 | 3 |
| 2 | MAPI_01 | Mashpishungo/Pambiliño | 0.1873 | -78.9128 | 653 | 21.6 | 99.8 | 2075 | leaf | 2 | 2 |
|  |  |  |  |  |  |  |  |  | necro | 5 | 3 |
|  |  |  |  |  |  |  |  |  | soil | 3 | 3 |
| 3 | MALO_01 | Mashpi Lodge | 0.1583 | -78.8819 | 827 | 19.4 | 99.6 | 2255 | leaf | 3 | 3 |
|  |  |  |  |  |  |  |  |  | soil | 4 | 4 |
| 4 | MALO_02 | Mashpi Lodge | 0.1685 | -78.8761 | 1018 | 20.6 | 99.7 | 2253 | leaf | 3 | 3 |
|  |  |  |  |  |  |  |  |  | soil | 3 | 3 |
| 5 | MIND_01 | Mindo Lindo | -0.0253 | -78.8129 | 1277 | 18.8 | 98.9 | 2704 | leaf | 4 | 4 |
|  |  |  |  |  |  |  |  |  | soil | 3 | 3 |
| 6 | RIBR_01 | Reserva Rio Bravo | -0.082 | -78.7353 | 1640 | 16.8 | 98.8 | 2347 | leaf | 12 | 12 |
|  |  |  |  |  |  |  |  |  | soil | 3 | 3 |
| 7 | INTI_02 | Reserva Intillacta | 0.0501 | -78.7219 | 1829 | 15.9 | 99.0 | 1939 | soil | 3 | 3 |
| 8 | INTI_01 | Reserva Intillacta | 0.0505 | -78.7232 | 1879 | 16.0 | 98.8 | 2076 | leaf | 1 | 1 |
|  |  |  |  |  |  |  |  |  | necro | 5 | 5 |
|  |  |  |  |  |  |  |  |  | soil | 3 | 3 |
| 9 | BECL_03 | Bellavista Cloud Forest | -0.0116 | -78.6893 | 2203 | 14.0 | 99.6 | 1595 | leaf | 8 | 8 |
|  |  |  |  |  |  |  |  |  | soil | 3 | 3 |
| 10 | CEDR_03 | El Cedral Ecolodge | 0.1132 | -78.5691 | 2212 | 14.3 | 98.6 | 1351 | leaf | 11 | 11 |
|  |  |  |  |  |  |  |  |  | necro | 5 | 5 |
|  |  |  |  |  |  |  |  |  | soil | 3 | 2 |
| 11 | BECL_02 | Bellavista Cloud Forest | -0.0124 | -78.6864 | 2282 | 14.3 | 99.1 | 1993 | soil | 3 | 3 |
| 12 | BECL_01 | Bellavista Cloud Forest | -0.0153 | -78.6863 | 2313 | 13.6 | 99.3 | 1595 | leaf | 8 | 8 |
|  |  |  |  |  |  |  |  |  | soil | 3 | 3 |
| 13 | CEDR_01 | El Cedral Ecolodge | 0.1195 | -78.5705 | 2492 | 13.0 | 97.7 | 1471 | leaf | 9 | 9 |
|  |  |  |  |  |  |  |  |  | soil | 3 | 3 |
| 14 | VERD_02 | Reserva Verdecocha | -0.1015 | -78.6004 | 2932 | 10.2 | 99 | 1251 | leaf | 4 | 4 |
|  |  |  |  |  |  |  |  |  | soil | 3 | 3 |
| 15 | VERD_03 | Reserva Verdecocha | -0.1044 | -78.6008 | 3109 | 9.9 | 96.2 | 1251 | leaf | 6 | 6 |
|  |  |  |  |  |  |  |  |  | soil | 3 | 3 |
| 16 | VERD_01 | Reserva Verdecocha | -0.1233 | -78.5958 | 3421 | 8.3 | 97 | 1271 | leaf | 8 | 7 |
|  |  |  |  |  |  |  |  |  | soil | 3 | 3 |
| 17 | YANA_01 | Reserva Yanacocha | -0.1267 | -78.5907 | 3507 | 7.2 | 98.7 | 1337 | leaf | 8 | 8 |
|  |  |  |  |  |  |  |  |  | soil | 3 | 3 |

595

**Table 2: Table containing the p-values of the Pearson's correlation coefficients depicted in Figure 4 and Figure 5. Per sample type (a,b,c), the correlated variables are: Altitude (ALT), temperature (MAT), humidity (HR), average chain length (ACL), ratio $C_{31}/(C_{31}+C_{29})$ (RATIO), carbon preference index (CPI), and the two axis of the nMDS analysis of each sample type (Axis 1, Axis 2).**

600

|        | MAT       | HR        | AP        | ACL       | RATIO     | CPI   | Axis 1    | Axis 2   |
|--------|-----------|-----------|-----------|-----------|-----------|-------|-----------|----------|
| ALT    | 0.000 *** | 0.000 *** | 0.000 *** | 0.000 *** | 0.000 *** | 0.100 | 0.000 *** | 0.506    |
| MAT    |           | 0.000 *** | 0.000 *** | 0.000 *** | 0.000 *** | 0.145 | 0.000 *** | 0.391    |
| HR     |           |           | 0.000 *** | 0.002 **  | 0.000 *** | 0.001 ** | 0.000 *** | 0.107  |
| AP     |           |           |           | 0.000 *** | 0.000 *** | 0.035 | 0.000 *** | 0.907    |
| ACL    |           |           |           |           | 0.000 *** | 0.272 | 0.000 *** | 0.001 ** |
| RATIO  |           |           |           |           |           | 0.611 | 0.000 *** | 0.531    |
| CPI    |           |           |           |           |           |       | 0.389     | 0.005 ** |
| Axis 1 |           |           |           |           |           |       |           | 1.000    |
| Axis 2 |           |           |           |           |           |       |           |          |

a) *leaf (86)*

|        | MAT       | HR        | AP        | ACL       | RATIO     | CPI   | Axis 1    | Axis 2 |
|--------|-----------|-----------|-----------|-----------|-----------|-------|-----------|--------|
| ALT    | 0.000 *** | 0.000 *** | 0.000 *** | 0.317     | 0.757     | 0.211 | 0.191     | 0.276  |
| MAT    |           | 0.000 *** | 0.000 *** | 0.304     | 0.764     | 0.222 | 0.186     | 0.256  |
| HR     |           |           | 0.000 *** | 0.328     | 0.752     | 0.203 | 0.197     | 0.295  |
| AP     |           |           |           | 0.157     | 0.923     | 0.591 | 0.141     | 0.051  |
| ACL    |           |           |           |           | 0.001 *** | 0.588 | 0.000 *** | 1.000  |
| RATIO  |           |           |           |           |           | 0.226 | 0.000 *** | 0.058  |
| CPI    |           |           |           |           |           |       | 0.426     | 0.224  |
| Axis 1 |           |           |           |           |           |       |           | 1.000  |
| Axis 2 |           |           |           |           |           |       |           |        |

b) *necromass (13)*

|        | MAT       | HR        | AP        | ACL       | RATIO     | CPI     | Axis 1    | Axis 2   |
|--------|-----------|-----------|-----------|-----------|-----------|---------|-----------|----------|
| ALT    | 0.000 *** | 0.000 *** | 0.000 *** | 0.000 *** | 0.001 **  | 0.004 ** | 0.000 *** | 0.897   |
| MAT    |           | 0.000 *** | 0.000 *** | 0.000 *** | 0.001 *** | 0.005 ** | 0.000 *** | 0.792   |
| HR     |           |           | 0.000 *** | 0.171     | 0.348     | 0.030   | 0.163     | 0.416    |
| AP     |           |           |           | 0.001 **  | 0.017     | 0.005   | 0.005 **  | 0.843    |
| ACL    |           |           |           |           | 0.000 *** | 0.907   | 0.000 *** | 0.164    |
| RATIO  |           |           |           |           |           | 0.420   | 0.000 *** | 0.513    |
| CPI    |           |           |           |           |           |         | 0.291     | 0.001 *** |
| Axis 1 |           |           |           |           |           |         |           | 1.000    |
| Axis 2 |           |           |           |           |           |         |           |          |

c) *soil (51)*

p-values

<.01 *, <0.005 **, <0.001 ***

**Figure 1: Map of the Pichincha transect. Dots and numbers refer to plots in Table 1, ordered by altitude. The green shading indicates altitude (m above sea level). The black line delineates the Pichincha project study area. Blue lines represent rivers, land codes as follows: COL = Colombia, ECU = Ecuador, PER = Perú.**

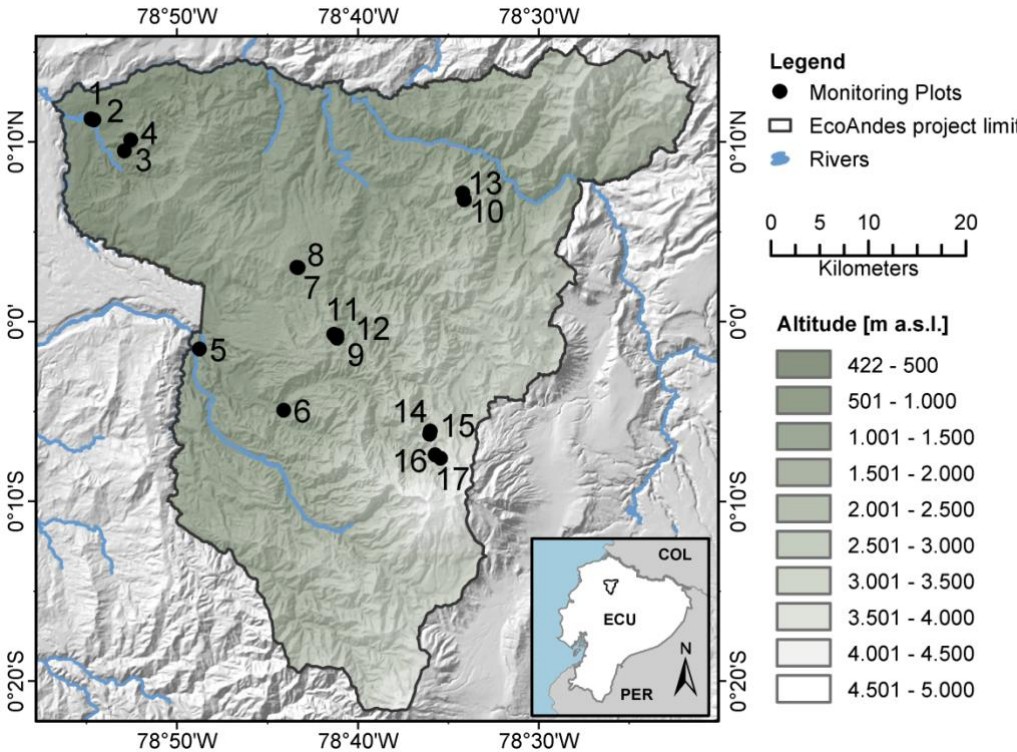

**Figure 2: Sample type average *n*-alkane distributions along the transect, showing average relative abundances (average % of total concentration (CONw)) across entire transect. Lines represent standard deviation from average, numbers in parentheses represent number of samples.**

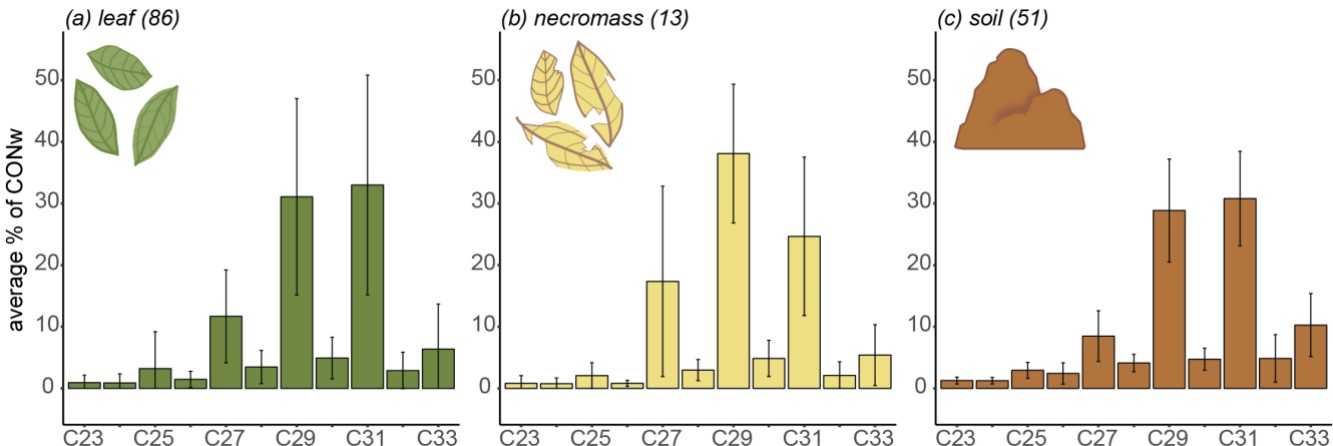

**Figure 3: nMDS plots of *n*-alkane patterns per sample type (a,b,c) and all sample types combined (d). Numbers in parentheses next to sample type represent number of samples. Arrows indicate fit of environmental variables, altitude (ALT), temperature (MAT), humidity (HR) and precipitation (AP). Text indicate *n*-alkane chain length contribution to sample. Samples indicated by symbols, sample type indicated by symbol colour and shape (green circles = leaf, yellow squares = necromass, brown diamonds = soil). Colour gradient in panels a, b and c indicate altitude (darker = lower, lighter = higher).**

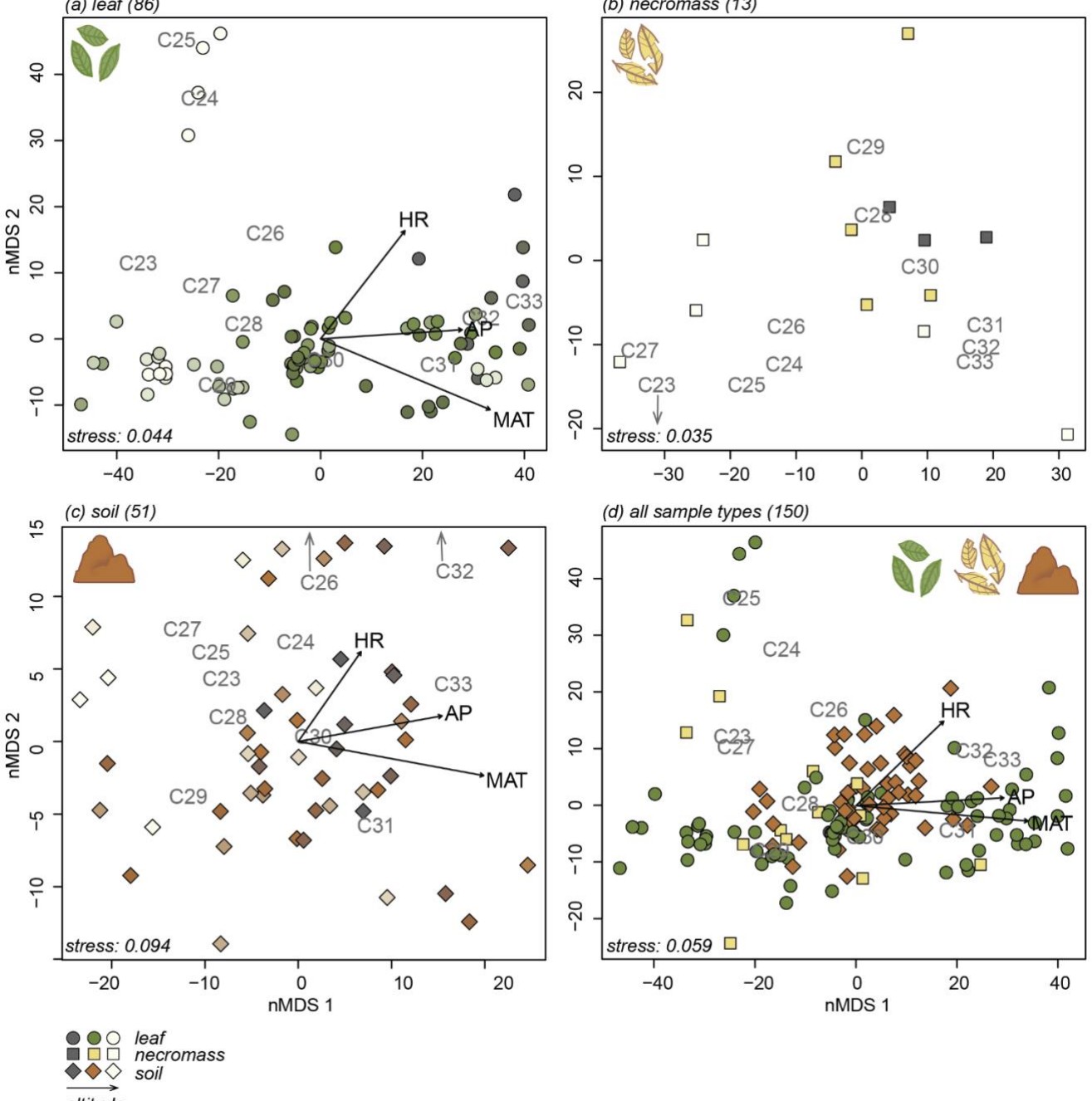

**Figure 4: Pearson's correlation coefficient matrix per sample type (a,b,c) showing the correlations between environmental variables (ALT = altitude, MAT = temperature, HR = humidity, AP = precipitation), *n*-alkane metrics (ACL = average chain length, RATIO = ratio C$_{31}$/(C$_{31}$+C$_{29}$), CPI = carbon preference index) and the nMDS axes (Axis 1, Axis 2). Numbers in parentheses represent number of samples. Significant correlations are circled, where the colour and circle size indicate the direction and correlation strength. Circle outlines reflect the significance level. Non-significant correlations are not circled.**

(a) leaf (86)

(b) necromass (13)

(c) soil (51)

p < .001 ———
p < .005 - - - -
p < .01 ............

−1  −0.8  −0.6  −0.4  −0.2  0  0.2  0.4  0.6  0.8  1

**Figure 5:** Scatterplots of *n*-alkane metrics (ACL = average chain length, RATIO = $C_{31}/(C_{31}+C_{29})$, CPI = carbon preference index) along the environmental gradient (MAT = temperature, HR = humidity, AP = precipitation). Symbols and colors indicate sample type (green circles = leaf, yellow squares = necromass, brown diamonds = soil), numbers in parentheses represent number of samples. Shading indicates the standard error of the linear correlation (lines), line type represents the significance level of the correlation. Represented correlation coefficients and p-values correspond to values in Figure 4 and Table 2.

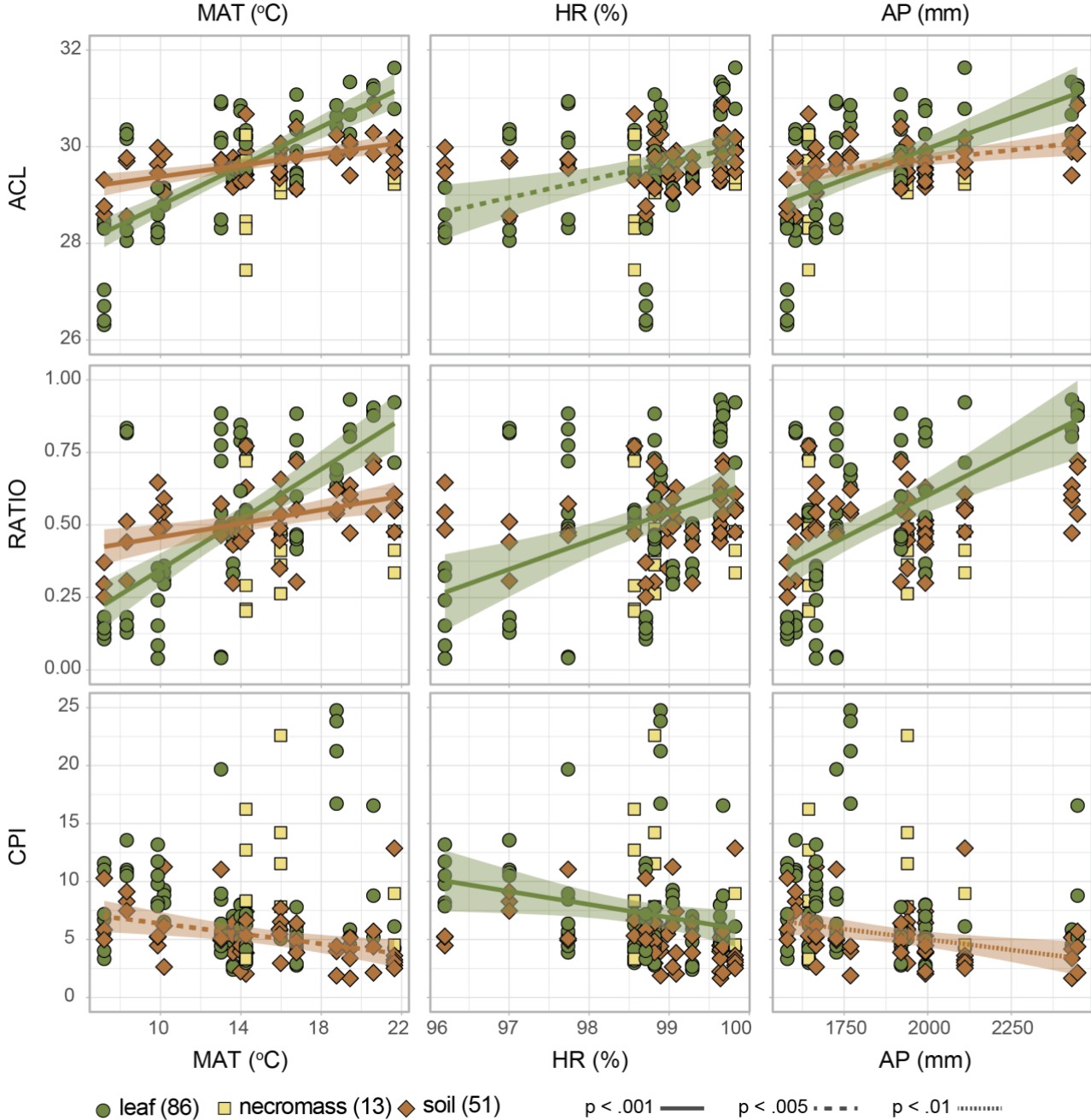