# Peer review of "From leaf to soil: *n*-alkane signal preservation, despite degradation along an environmental gradient in the tropical Andes."

_Biogeosciences, 2019_

## Referee Comment (RC1) · Anonymous Referee #1 · 3 Feb 2020

The draft untitled "From leaf to soil: n-alkane signal preservation, despite degradation along an environmental gradient in the tropical Andes" reports the evolution of n-alkane signature in leaves, litter and soil organic matter along an environmental gradient. The study has been properly conducted and represents an extensive amount of work. The draft is clear and well-written. However, I have several comments which may greatly diminish the implications of this work regarding the potential utilization of n-alkane biomarkers for the reconstruction of past environmental changes.

1° Several papers reported that soil organic carbon is mostly derived from roots (e.g. Rasse et al., 2005). It would have been more sounded to analyze n-alkane in roots

instead of leaves.

2° On top of that, n-alkanes in soil organic matter derive from actual vegetation but also from past vegetation. We therefore do not know which plants these alkanes come from and in which climate the plants which produced these alkanes have grown. To this respect, I do not see the rationale for comparing alkanes in soil organic matter and leaves, especially in the context of reconstructing past environmental changes.

3° We can see on Figure 5, that if n-alkane signature in soil organic matter is some-time significantly related to actual climate conditions, the slopes are low and the re-lationships are scattered. It means that the predictions that can be made from these relationships would likely be very uncertain. I don't know if it would be informative for paleoclimatologists to know that the MAT 15°C +/- 15°C (95% confidence interval).

I suggest that the authors take my comment into account to discuss their nice dataset and tune down the implications that their study can have for reconstructing past environmental conditions.

Reference cited : Rasse DP, Rumpel C, Dignac MF (2005) Is soil carbon mostly root carbon? Plant & Soil, 269:341-356.

---

## Referee Comment (RC2) · Anonymous Referee #2 · 26 Feb 2020

General comments: The manuscript by Teunissen van Manen et al. aims to improve the understanding of the mechanisms controlling long-chain n-alkane distributions preserved in sedimentary archives. The study analyzes how long-chain n-alkane distributions change in response to incorporation into soils and how environmental parameters such as temperature and precipitation influence alkane distributions. To meet these goals the authors presents long-chain n-alkane data from leaves, necro mass and soils along an altitude transect in the Ecuadorian Andes. The authors conclude that the ACL of long-chain n-alkanes does not systematically change among the three studied n-alkane pools, but that the variability of ACL among soil samples is lower than in leaf and necro mass samples. They also infer a dominant influence of temperature

on ACL. While the authors present a novel potentially interesting data set representing considerable analytical effort, there are several major issues that need to be addressed and revised. Since the environmental parameters discussed in the manuscript are all correlated to altitude, it is hard to tell which of the parameters control the ACL of alkanes. The authors mention precipitation, temperature and humidity (whereas this parameter barely changes over the transect), but they do not discuss other factors along the altitude transect that could also play a role (e.g. lower atmospheric pressure and thereby pCO2). So the study setup is not ideal to disentangle the environmental factors controlling ACL. The manuscript unfortunately overstates the novelty of its results and does not properly acknowledge previous findings. The authors should carefully review the available literature and to work out which aspects of the study offer new insights and where it contrasts to previous studies. Leaf-litter experiments and previous soil studies for instance showed a shift of ACL during degradation in necro mass and soils (e.g. Wu et al. 2019, OG, Zech et al. 2011, GCA). So it would be worthwhile to discuss why there is no such change in the transect analyzed in this manuscript, while other studies indeed find an impact.

Specific comments: Lines 13-14, 52-54, 257-260: The contention that very little is known about degradation processes altering long-chain n-alkane distributions is not correct and there is ample literature on this issue. For instance, Zech et al. 2011, GCA have systematically studied these processes in litter bag experiments. Schäfer et al. 2016, SOIL studied shifts in long-chain n-alkane distributions between litter and soil samples in a transect across Europe. All of this is unfortunately not discussed in the manuscript. The authors also contend that previous work on soils and necro mass is limited to the four publications given in line 54, which should be expanded.

Lines 32, 33; 38. To call long-chain n-alkanes a new proxy is not correct. The classical paper on leaf-wax n-alkanes by Eglinton and Hamilton 1967 in Science has been published more than five decades ago. Application of n-alkanes as environmental proxy also dates back decades (see for instance Huang et al. 2000, GCA, Rieley et al. 1991,

[Figure]

Nature,).

Lines 38-40. The authors claim that the relative distribution of n-alkanes can be used to reconstruct past precipitation and temperature. The cited studies show indeed relations between chain-length distributions and temperature or precipitation. But Hoffmann et al. 2013 OG highlight that there is a multitude of factors influencing ACL values and do not suggest to use this as a specific proxy. Other studies have also highlighted the impact of vegetation on ACL (e.g. Rommerskirchen et al. 2006 OG). So the factors influencing ACL are diverse and regionally different, which limits its application as proxy.

Lines 83-89: As one of the altitude induced gradients for their analysis, the authors mention relative humidity. Yet humidity only varies between 96 and 99% and is therefore always close to 100%. So, this parameter is maxed out over the entire gradient and does not show any pronounced variability and is therefore hardly suited to explain variability in alkane distributions.

Lines 203-205 and Figure 3: The authors claim that the long-chain n-alkane distribution among the different sample types does not show a significant difference upon visual inspection of their nMDS plots. It would be great if the authors confirm this finding by applying statistical analysis (e.g. t-test).

Lines 369-370: To conclude from this study that even ancient alkane distributions in sediments are remain constant is a little far fetched The most degraded pool this study analyses are soils. To claim that ancient distributions are constant one would also need to study transport (e.g. aeolian or riverine) to the deposition area and the effects of diagenesis thereafter. In some cases the distributions might indeed be constant, but there are changes during transport, and in response to thermal overprint that have been reported in the past (e.g. Wang et al. 2017, OG) and need to be considered.

Lines 375-378: The authors claim that ACL and to a lesser degree also CPI reflects temperature. Given the large variability shown in Fig. 5 this is again an overstatement.

Since there is a large number of environmental parameters that follow the altitude gradient it is unclear if it is really temperature that controls ACL. If this was indeed the case, the large variability in the regression precludes meaningful statements on past temperature from sediment records.

Technical corrections: Line 12: The word "however" is redundant here and can be deleted.

Lines 15-18: This sentence is convoluted. It is also unclear what the difference between "n-alkane pattern, the n-alkane signal, and the local environmental information reflected in the n-alkane signal" is. Rephrase the sentence and be more specific in your terminology. Later in line 56 more explanation is offered which should be moved to the first occurrence in the text.

Line 22: Is "the primary observed n-alkane signal" the alkane signal observed in the leaf samples. If so, mention this specifically.

Lines 22 and 24: What kind of parameter is meant by "the environment". Be more specific.

Line 30: As the sentence refers to reconstruction replace soils with paleosols.

Line 66: The word altitudinally is misspelt.

Line 84: Rephrase the second part of the sentence to "including temperature, humidity and precipitation."

Line 99: Add a space after the number 15.

Line 159-160: It should be mentioned explicitly that the ACL only features compounds of odd carbon chain length.

Line 165: The word "the" before Marzi et al. can be removed.

Line 177: This sentence is on statistical convention. So there are probably better

references available to support this.

Lines 207: What ratio metrics are meant here? Be more specific.

Line 229: Consider replacing the word complete with perfect.

Lines 273-274: As is, this sentence reads as if there is no odd over even pattern in the Leaf and necro mass samples and should therefore be rephrased.

Line 374: Specify that the leaf wax patterns remain stable.

Appendix B1: The axis labels are hard to read and should be larger or thicker.

---

## Author Comment (AC2) · 25 Mar 2020

We thank Reviewer 2 for their recognition of our novel dataset and constructive comments that have helped us to develop our manuscript further. We respond to the general points raised, and then specific issues, below:

General comments

1. Reviewer 2 expresses concerns regarding our ability to disentangle environmental factors controlling the n-alkane pattern shifts given the extent of the environmental gradients studied; particularly with regard to: (a) the limited gradient in relative air

humidity, and (b) a lack of consideration of pCO2.

a) We accept the reviewer comment on the gradient range. However, we do not feel that this compromises the integrity of our manuscript, because the focus on the manuscript is on the degradation process, not the environmental gradient. We address the entanglement of the environmental variables specifically in lines 232-235, and agree that disentangling them is not possible in this study. We will address the implications of the environmental entanglement more explicitly in the discussion section. See also our response under reviewer comment "Lines 83-89".

b) We did not consider the gradient in pCO2, as there is evidence for a link between pCO2 and n-alkanes isotope signatures (via C3/C4 plant distributions)(e.g. Boom et al., PALAEO3, 2002), but there is no evidence for a link between pCO2 and n-alkane patterns in literature. However, there is strong evidence for a link between temperature, humidity and precipitation and n-alkane patterns (e.g. Bush and McInerney, OG, 2015; Hoffmann et al., OG, 2013; Tipple and Pagani, GCA, 2013), which is why we chose to include those variables. Additionally, our transect spans across a montane forest, where C3 plants dominate along the whole gradient. We do not see it fitting to include a discussion on pCO2 in this manuscript, but will address our reasoning for choosing the particular environmental gradient more explicitly.

2. Reviewer 2 expresses concerns regarding the overstatement of the novelty of the findings and that our findings are not sufficiently discussed and/or embedded in relevant literature. We thank the reviewer for providing suggestions of additional literature. We will include the missing literature reference provided by the reviewer (Schäfer et al, 2016 SOIL) and review our wording in the discussion and conclusion sections to not suggest novelty where it is not applicable. Specifically, Reviewer 2 suggests to include a discussion of why our results "do not show a shift in ACL during degradation in necromass and soils", in contrast to what has been found in other studies (Wu et al., OG, 2019, Zech et al., GCA, 2011). We are unsure what the reviewer means by a "shift in ACL during degradation", as our study does not track degradation over time as is

done by Zech et al. (GCA, 2011). In the scope of our study, we interpret the comment to mean "discuss why the ACL of necromass and soil n-alkanes do not shift along the gradient". We do not see a necromass ACL shift along the gradient due to the limited sampling along the gradient (lines 243-244). We do see a change in soil ACL along the gradient, which we have discussed in light of Wu et al (2019) (lines 302-303).

Specific comments

Lines 13-14 52-54, 257-260: We accept the reviewer comment. The aim of our statements was to indicate that compared to the large number of leaf wax n-alkane distribution studies, necromass and soil n-alkane patterns studied are limited. Especially in the context of the n-alkane patterns palaeoecological proxy literature, we find there is a gap of knowledge in the taphonomic processes that influence the interpretation of the proxy. Additionally, we find a limited number of studies that focus on n-alkane patterns (rather than quantities and isotopes) (such as Wu et al., OG, 2019) and are set in the 'natural' tropical settings (rather than temperate agricultural settings)(such as Wiesenberg et al., 2004; Zech et al., GCA, 2011). We appreciate the missing literature reference of Schafer et al. (2016, SOIL), but we do think it is fair to state there is a knowledge gap in our understanding of how taphonomic processes complicate/alter the interpretation of the n-alkane patterns proxy (in the tropics in particular). We will alter the wording to avoid any possible suggestion that we claim there is no available literature.

Line 54: the sentence lists studies that study and compare both plant material *and* soils, not all studies that have studied necromass *or* soils. We will reword the sentence to avoid confusion.

Lines 32, 33, 38: We are of course aware of the fact that extractable lipids in general, and n-alkanes in particular have been studied for decades (see e.g. a review article on this by one of the co-authors: Jansen & Wiesenberg, SOIL, 2017). However, the application of n-alkanes as palaeoecological proxy with a more detailed interpretation of the

signal than simply as indicator of input of terrestrial higher plant material is something that has only gained ground in the last 15 years. Still, even such application until now has been mainly focused on reconstructing past vegetation distributions, or via isotope signatures, past climatic reconstructions. The use of the chain length distribution patterns of n-alkanes as proxy for climatic conditions is indeed novel and, when further tested and developed, would add a valuable tool to the previously mentioned existing palaeoecological application of n-alkanes and/or their isotopic signatures. It is clear we failed to properly specify the novelty of the application we discussed in our manuscript. We will alter the statement so it is clear we mean to say that the application of n-alkane patterns as a palaeoecological proxy (not isotopes or other applications) are relatively new and under development.

Lines 38-40: We accept the reviewer comment. The sentence aimed to lists previous findings that suggest n-alkane patterns are a promising development in palaeoecological proxies. We will reword the sentence so we are clear about what aspect of the n-alkanes proxy we consider novel and promising (also see previous comment).

Lines 83-89: We accept the reviewer comment. Although the humidity gradient is short and close to 100% along much of the gradient, we chose to include the variable because we want to include all available environmental variables that have been found to relate to n-alkane distribution changes in previous studies (also see Reviewer 1 comment 2). Additionally, although the humidity gradient range of variation is only narrow, excluding it would introduce noise in the interpretation of the other correlations. We agree the limitations of the humidity gradient need to be discussed explicitly. We will include a statement acknowledging the limitations of the humidity gradient and the implications this has for the results on line 234.

Lines 203-205 and Figure 3: We disagree with the reviewer comment. We think that the nMDS of the sample types combined (Fig 3d) shows unambiguous overlap between the sample types, to the extent that we do not see added value in performing additional statistical tests to show that the sample types have similar n-alkane patterns

(as nMDS is a widely accepted statistical analysis to show (dis)similarity between samples). However, we acknowledge that the wording suggests statistical testing, and will change the wording to better reflect the analysis performed.

Lines 369-370: We do not think that the sentence "Taken together, our results and previous findings [. . .] suggest that ancient n-alkane signals likely carry environmental information similar to that observed in modern leaves, necromass and soils" claims (or suggests) (a) that sedimentary n-alkanes are constant or (b) that no other processes affect the interpretation of sedimentary n-alkanes. Additionally, the next sentence acknowledges the existence of other factors that complicate the interpretation of the sedimentary n-alkane record. We agree with the reviewer that it would be a stretch to claim that modern n-alkanes directly translate to sedimentary n-alkanes. We do not see it necessary to change the original sentence, but we will elaborate and include the factors suggested by the reviewer (transport and thermal overprinting) to the existing caveat sentence as additional examples of complicating factors.

Lines 375-378: We accept the reviewer comment. There is indeed large scatter in the correlations in Figure 5, in particular in the leaf sample data. On the particular line we will change "reflecting" to "correlate with" so it is clear we do not suggest the correlation is perfect. We will also include a statement that acknowledges the variance in these correlations, and what this implies for the palaeoecological proxy (section 4.3).

Technical corrections All technical corrections are accepted and will be incorporated in the manuscript.

---

## Author Response (AR1)

Response to Editor                                          14 September 2020

Dear Sébastien Fontaine,

We agree with the reviewers notes that leaf input and the correlative methods used limit our study interpretation and application. Accordingly, we have altered our approach in responding to those comments to ensure they are fully addressed. You will find dedicated sections addressing root input, $pCO_2$, the correlative nature of our results and the limitations of the application of our findings. We now feel confident the manuscript presents the novelty of the findings within the scope of the study. We present you with the updated manuscript, a list of relevant changes, and an updated point-by-point review responses document.

Best regards,
Milan Teunissen van Manen

List of relevant changes

- Rewrote abstract to reflect changes made to manuscript
- Expanded introduction so clearly states the manuscript focuses on the novel application of n-alkane patterns, rather than n-alkane biomarkers as a whole.
- Minor wording alterations in the methods section
- The discussion now includes a paragraph on root n-alkane input to soil samples and addresses potential root n-alkane input wherever relevant.
- The discussion now includes a paragraph on $pCO_2$
- The discussion now includes addresses the correlative nature and noise of our study.
- The discussion section now includes a paragraph where we address the limitations and contextualize the proposed applications of our results.
- Rewrote the conclusions section to reflect tone of the manuscript.

Response to reviewer comments on the Biogeosciences Discussions.

Reviewer 1

We are grateful to Reviewer 1 for their positive and helpful comments that have helped with the development of our manuscript. Below we respond directly to each of the suggestions in turn:

1.  Reviewer 1 notes that roots make up a significant proportion of organic carbon found in soils, and asks if the study should have included the analysis of roots. We accept the reviewer comment. One of the aims of the paper is to study whether the discoveries made in plant wax *n*-alkanes (for example, Bush and McInerney, GCA, 2013; Feakins et al., OG, 2016) are also reflected in soils (a more degraded stage of *n*-alkane substrate). For this reason, we chose to compare the *n*-alkane patterns from leaves to necromass and soil samples in this study. We agree root input of *n*-alkanes can be substantial in soils, however, we wish to note that the degree to which this influences the overall *n*-alkane pattern observed is subject to ongoing scientific debate. The article cited by the reviewer (Rasse et al., P&S, 2005) focuses on the stabilization of bulk soil organic matter, rather than the specific *n*-alkane fraction. The bulk soil organic matter need not, and based on present scientific insights (e.g. Lehmann & Kleber, Nature, 2015) does not, reflect the same distribution of origin as the *n*-alkane fraction. For instance, while root input may be important for bulk organic matter, the amount of *n*-alkanes produced by roots is usually much lower than by leaves (e.g. Jansen & Wiesenberg, SOIL, 2017). Furthermore, if root input dominated the *n*-alkane patterns in the soil in our study, we would expect a different *n*-alkane pattern in the soils when compared with the *n*-alkane patterns of the leaves and necromass (Jansen et al., OG, 2006). However, we do realize that the rationale behind comparing n-alkanes from leaves, necromass and soils was not clear in our initial version of the manuscript and the root source should be addressed. *We have strengthened the wording of the rationale/aim and included a paragraph that outlines the root n-alkane input to soil samples discussion as presented above.*

2. Reviewer 1 expresses concerns with the rationale behind comparing leaf n-alkanes to soil n-alkanes, considering modern soils contain both modern and ancient organic matter, and that ancient organic matter may not derive from the same plants as those found at the site today. We agree with the reviewer that soils contain organic material of varying ages and potentially incorporate material from plant assemblage different to the current one. However, we expect that the majority of the organic material in our soil samples was derived from the modern vegetation, because our soils samples were derived from the upper 5 cm of the soil (immediately below the leaf litter layer). We are therefore confident that our soil samples represent organic material derived from vegetation similar to modern. Regardless, knowing the exact source vegetation of the n-alkanes is not relevant in this study, as the aim of our manuscript is to explore whether the *n*-alkane patterns degrade, as the substrate degrades (rather than whether the parent vegetation *n*-alkane patterns are reflected in the soil *n*-alkane patterns). See also our rationale in the previous comment. Based on our findings we argue that the relationship between plant and soil *n*-alkanes is unlikely to be direct, that it seems likely that soil specific processes such as microbial reworking, source mixing, and spatiotemporal averaging make it hard to link modern plant wax *n*-alkane knowledge to soil or sedimentary *n*-alkane knowledge (and thus agree with the reviewer comment). *We recognize that the manuscript rationale, aim, and conclusion can be featured more prominently. We altered the introduction so it is clear why we compare the n-alkane patterns leaves, necromass and soil samples.*

3. Reviewer 1 notes that there is uncertainty in the relationship between the *n*-alkane signature and climate. We agree with Reviewer 1's observations of Figure 5 that the scatter is high; however, we feel that our interpretation of the data is valid because we do not link the magnitude of the shifts in *n*-alkane patterns to reflect magnitude of change (such as a specific temperature range), but rather focus on the direction of change. *We have dedicated a paragraph to the text noting that our results should be seen as a proof-of-principle of a new proxy focusing on a qualitative assessment of the direction of change over time that still needs further development. We have now also explicitly state discuss the uncertainty in these correlations, and what the study*

*limitations imply for the further development the palaeoecological proxy (section 4.3).*

Reviewer 2

We thank Reviewer 2 for their recognition of our novel dataset and constructive comments that have helped us to develop our manuscript further. We respond to the general points raised, and then specific issues, below:

General comments

1. Reviewer 2 expresses concerns regarding our ability to disentangle environmental factors controlling the *n*-alkane pattern shifts given the extent of the environmental gradients studied; particularly with regard to: (a) the limited gradient in relative air humidity, and (b) a lack of consideration of pCO2.

   a) We accept the reviewer comment on the gradient range. However, we do not feel that this compromises the integrity of our manuscript, because the focus on the manuscript is on the degradation process, not the environmental gradient. We address the entanglement of the environmental variables specifically in lines 232-235, and agree that disentangling them is not possible in this study. *We will address the implications of the environmental entanglement more explicitly in the discussion section. See also our response under reviewer comment "Lines 83-89".*

   b) We did not consider the gradient in pCO2, as there is evidence for a link between pCO2 and *n*-alkanes isotope signatures (via C3/C4 plant distributions)(e.g. Boom et al., PALAEO3, 2002), but there is no evidence for a link between pCO2 and *n*-alkane patterns in literature. However, there is strong evidence for a link between temperature, humidity and precipitation and *n*-alkane patterns (e.g. Bush and McInerney, OG, 2015; Hoffmann et al., OG, 2013; Tipple and Pagani, GCA, 2013), which is why we chose to include those variables. Additionally, our transect spans across a montane forest, where C3 plants dominate along the whole gradient. *We have included a discussion of $pCO_2$ in the manuscript and addressed our reasoning for choosing the particular environmental gradient more explicitly.*

2. Reviewer 2 expresses concerns regarding the overstatement of the novelty of the findings and that our findings are not sufficiently discussed and/or embedded in relevant literature. We thank the reviewer for providing suggestions of additional literature. *We have included the missing literature reference provided by the reviewer (Schäfer et al, 2016 SOIL) and reviewed our wording in the discussion and conclusion sections to not suggest novelty where it is not applicable.*

   - Specifically, Reviewer 2 suggests to include a discussion of why our results "do not show a shift in ACL during degradation in necromass and soils", in contrast to what has been found in other studies (Wu et al., OG, 2019, Zech et al., GCA, 2011). We are unsure what the reviewer means by a "shift in ACL during degradation", as our study does not track degradation over time as is done by Zech et al. (GCA, 2011). In the scope of our study, we interpret the comment to mean "discuss why the ACL of necromass and soil *n*-alkanes do not shift along the gradient". We do not see a necromass ACL shift along the gradient due to the limited sampling along the gradient (lines 243-244). We do see a change in soil ACL along the gradient, which we have discussed in light of Wu et al (2019) (lines 302-303).

Specific comments

Lines 13-14 52-54, 257-260: We accept the reviewer comment. The aim of our statements was to indicate that compared to the large number of leaf wax *n*-alkane distribution studies, necromass and soil *n*-alkane patterns studied are limited. Especially in the context of the *n*-alkane patterns palaeoecological proxy literature, we find there is a gap of knowledge in the taphonomic processes that influence the interpretation of the proxy. Additionally, we find a limited number of studies that focus on *n*-alkane patterns (rather than quantities and isotopes) (such as Wu et al., OG, 2019) and are set in the 'natural' tropical settings (rather than temperate agricultural settings)(such as Wiesenberg et al., 2004; Zech et al., GCA, 2011). We appreciate the missing literature reference of Schafer et al. (2016, SOIL), but we do think it is fair to state there is a knowledge gap in our understanding of how taphonomic processes complicate/alter the interpretation of the *n*-alkane patterns proxy (in the tropics

in particular). *We have altered the section wording to avoid any possible suggestion that we claim there is no available literature.*

Line 54: the sentence lists studies that study and compare both plant material **and** soils, not all studies that have studied necromass **or** soils. *We have reworded the paragraph wording to avoid confusion.*

Lines 32, 33, 38: We are of course aware of the fact that extractable lipids in general, and *n*-alkanes in particular have been studied for decades (see e.g. a review article on this by one of the co-authors: Jansen & Wiesenberg, SOIL, 2017). However, the application of *n*-alkanes as palaeoecological proxy with a more detailed interpretation of the signal than simply as indicator of input of terrestrial higher plant material is something that has only gained ground in the last 15 years. Still, even such application until now has been mainly focused on reconstructing past vegetation distributions, or via isotope signatures, past climatic reconstructions. The use of the chain length distribution patterns of *n*-alkanes as proxy for climatic conditions is indeed novel and, when further tested and developed, would add a valuable tool to the previously mentioned existing palaeoecological application of *n*-alkanes and/or their isotopic signatures. It is clear we failed to properly specify the novelty of the application we discussed in our manuscript. *We have altered the introduction section so it is clear we mean to say that the application of n-alkane patterns as a palaeoecological proxy (not isotopes or other applications) are relatively new and under development.*

Lines 38-40: We accept the reviewer comment. The sentence aimed to lists previous findings that suggest *n*-alkane patterns are a promising development in palaeoecological proxies. *We have altered the wording so we it is clear what aspect of n-alkanes proxy we consider novel and promising (also see previous comment).*

Lines 83-89: We accept the reviewer comment. Although the humidity gradient is short and close to 100% along much of the gradient, we chose to include the variable because we want to include all available environmental variables that have been found to relate to *n*-alkane distribution changes in previous studies (also see Reviewer 1 comment 2).

Additionally, although the humidity gradient range of variation is only narrow, excluding it would introduce noise in the interpretation of the other correlations. *We have included a statement acknowledging the limitations of the humidity gradient and the implications this has for the results.*

Lines 203-205 and Figure 3: We disagree with the reviewer comment. We think that the nMDS of the sample types combined (Fig 3d) shows unambiguous overlap between the sample types, to the extent that we do not see added value in performing statistical tests to show that the sample types have similar *n*-alkane patterns (as nMDS is a widely accepted statistical analysis to show (dis)similarity between samples). *However, we acknowledge that the wording suggests statistical testing, and have changed the wording to better explain this.*

Lines 369-370: We do not think that the sentence "Taken together, our results and previous findings […] suggest that ancient n-alkane signals likely carry environmental information similar to that observed in modern leaves, necromass and soils" claims (or suggests) (a) that sedimentary n-alkanes are constant or (b) that no other processes affect the interpretation of sedimentary n-alkanes. Additionally, the next sentence acknowledges the existence of other factors that complicate the interpretation of the sedimentary n-alkane record. We agree with the reviewer that it would be a stretch to claim that modern n-alkanes directly translate to sedimentary n-alkanes. *We have reworded the discussion section where we address the implications of our findings more extensively and explicitly discuss appropriate caveats.*

Lines 375-378: We accept the reviewer comment. There is indeed large scatter in the correlations in Figure 5, in particular in the leaf sample data. *We have included a section that acknowledges the variance in these correlations, and what this implies for the palaeoecological proxy (section 4.3).*

Technical corrections

All technical corrections are accepted and will be incorporated in the manuscript.

[revised manuscript text omitted]